# Mutual Information Collapse Explains Disentanglement Failure in $\beta$-VAEs

**Minh Vu**                                            *vumh2@miamioh.edu*
*Department of Mathematics*
*Miami University*

**Xiaoliang Wan**                                      *xlwan@lsu.edu*
*Department of Mathematics*
*Louisiana State University*

**Shuangqing Wei**                                     *swei@lsu.edu*
*Division of Electrical & Computer Engineering*
*Louisiana State University*

**Reviewed on OpenReview:** *https://openreview.net/forum?id=POE4LsRvGY*

## Abstract

The $\beta$-VAE is a foundational framework for unsupervised disentanglement, utilizing the regularization parameter $\beta$ to balance latent factorization against reconstruction fidelity. However, disentanglement performance often exhibits a non-monotonic dependence on $\beta$: standard metrics, such as MIG and SAP, typically peak at intermediate values and deteriorate under stronger regularization. We characterize this phenomenon as informational collapse—an information-theoretic failure in which excessive regularization drives the mutual information between latent variables and ground-truth generative factors toward zero. By analyzing the stationarity conditions in a linear-Gaussian setting, we prove that, for $\beta > 1$, alternating optimization induces a spectral contraction of the encoder gain, causing its spectral norm to decay exponentially and latent–factor mutual information to vanish. Motivated by this failure mode, we investigate the $\lambda\beta$-VAE, which augments the objective with an auxiliary $L_2$ reconstruction penalty. Our analysis shows that this term modifies the encoder stationarity conditions and weakens the derived upper bound on encoder-gain contraction relative to the standard $\beta$-VAE. Experiments on dSprites, Shapes3D, and MPI3D-real indicate that configurations with $\lambda > 0$ generally improve reconstruction accuracy and exhibit an attenuated decline in disentanglement performance relative to the $\lambda = 0$ baseline, particularly under strong regularization.

## 1 Introduction

Disentangled representation learning aims to map high-dimensional observations into a latent space in which individual dimensions correspond to distinct, often statistically independent, generative factors (Bengio et al., 2014; Tschannen et al., 2018; Wang et al., 2024). This structural alignment is critical for interpretability in scientific domains—such as genomics, robotics, and physics—where latent variables are expected to reflect underlying physical or biological mechanisms (Eguchi et al., 2022; Jiao et al., 2024; Cropsal & Mercado, 2025). Variational autoencoders (VAEs) provide a principled probabilistic framework for learning such representations (Kingma & Welling, 2022), with the $\beta$-VAE (Higgins et al., 2017) serving as a canonical baseline. By upweighting the Kullback–Leibler (KL) divergence term in the evidence lower bound (ELBO), the $\beta$-VAE imposes an information bottleneck that promotes latent factorization and enables unsupervised disentanglement under suitable conditions.

However, the $\beta$-VAE exhibits a well-documented failure mode under strong regularization. One related phenomenon is *posterior collapse* (He et al., 2019; Lucas et al., 2019; Razavi et al., 2019; Wang & Ziyin, 2022; Ichikawa & Hukushima, 2024; 2025), in which the approximate posterior approaches the prior, rendering the latent variables uninformative about the observations. In this work, we adopt a broader perspective and study the degradation of factor-relevant latent information, which we refer to as *informational collapse*. Empirically, disentanglement performance—measured by metrics such as the Mutual Information Gap (MIG) (Chen et al., 2018) and Separated Attribute Predictability (SAP) (Kumar et al., 2018)—typically depends non-monotonically on $\beta$: performance improves at moderate values but deteriorates as $\beta$ increases (Locatello et al., 2019; Schott et al., 2022). Although this trend is often attributed to a trade-off between reconstruction fidelity and latent factorization, a mechanistic explanation for the loss of factor-relevant information under strong regularization remains limited.

We argue that the utility of disentanglement metrics fundamentally depends on *latent informativeness*—the extent to which latent variables retain information about the ground-truth generative factors. When latent–factor mutual information becomes negligible, metrics such as MIG and SAP lose discriminative power and can no longer meaningfully distinguish structured representations from uninformative ones. Understanding this regime is essential for diagnosing instability in disentanglement and for addressing broader challenges in dimensionality reduction, where avoiding the collapse of learned representations is a primary concern (Kalantidis et al., 2022; Baptista et al., 2025).

To address this gap, we provide a formal analysis of $\beta$-VAE optimization dynamics within a tractable linear-Gaussian setting. We prove that, for $\beta > 1$, alternating optimization induces a *spectral contraction* of the encoder gain, causing its spectral norm to decay exponentially and driving the iterates toward an uninformative trivial solution in which latent–factor mutual information vanishes. Motivated by this failure mode, we investigate the $\lambda\beta$-VAE (Vu, 2025), which augments the objective with an auxiliary $L_2$ reconstruction penalty weighted by $\lambda$. Within the same framework, we show that this term modifies the encoder stationarity conditions and weakens the derived upper bound on encoder-gain contraction relative to the standard $\beta$-VAE. Rather than guaranteeing the preservation of latent informativeness, this result characterizes how the auxiliary penalty changes the condition under which the derived bound establishes contraction. We further examine whether similar qualitative trends arise empirically in deep nonlinear VAEs. Experiments on dSprites (Matthey et al., 2017), Shapes3D (Burgess & Kim, 2018), and MPI3D-real (Gondal et al., 2019) indicate that configurations with $\lambda > 0$ generally achieve lower reconstruction error and higher MIG and SAP scores than the $\lambda = 0$ baseline, particularly under strong regularization. These findings characterize the empirical behavior of deep convolutional architectures and should not be interpreted as a direct validation or extension of the linear-Gaussian theorem.

**Summary of Contributions**

- **Mechanism of Informational Collapse:** Within a linear-Gaussian framework and under the specified alternating-optimization procedure, we establish spectral contraction of the encoder gain for $\beta > 1$. This contraction drives its spectral norm and latent–factor mutual information toward zero, yielding an uninformative trivial solution.
- **Failure Mode of Disentanglement Metrics:** We demonstrate that standard disentanglement metrics, such as MIG and SAP, lose discriminative power as latent–factor mutual information vanishes. This failure mode may help explain the metric reliability and reproducibility challenges reported in prior work. (Abdi et al., 2019; Fil et al., 2021; Carbonneau et al., 2024).
- **Analysis of the $\lambda\beta$-VAE Framework:** We derive the encoder stationarity conditions for the $\lambda\beta$-VAE and show that the $\lambda$-weighted $L_2$ reconstruction penalty weakens the derived upper bound on encoder-gain contraction relative to the standard $\beta$-VAE. This analysis clarifies how $\lambda$ changes the contraction condition without guaranteeing non-vanishing latent informativeness.
- **Empirical Evaluation:** We examine whether related qualitative trends arise on dSprites, Shapes3D, and MPI3D-real. Configurations with $\lambda > 0$ generally achieve lower reconstruction error and higher MIG and SAP scores than the $\lambda = 0$ baseline, particularly under strong regularization.

## 2 Related Work

**VAE Regularization and Trade-offs.** The $\beta$-VAE (Higgins et al., 2017) leverages an upweighted KL divergence to foster latent factorization by constraining the capacity of the latent information bottleneck. Building on this formulation, $\beta$-TCVAE (Chen et al., 2018) and FactorVAE (Kim & Mnih, 2018) decompose the ELBO to isolate and penalize the total correlation of the aggregated posterior, whereas DIP-VAE (Kumar et al., 2018) enforces disentanglement via moment-matching. Further extensions explore alternative latent structures, such as discrete variables via stochastic quantization (Takida et al., 2022) or hyperbolic geometries for hierarchical representations (Cho et al., 2023). Collectively, these methods promote latent independence by imposing structural constraints on the latent distribution. However, empirical studies indicate that strong regularization can reduce latent informativeness and produce unstable disentanglement performance in high-regularization regimes (Fil et al., 2021). These observations highlight a fundamental trade-off between promoting latent independence and retaining factor-relevant information.

**Challenges in Unsupervised Disentanglement.** Unsupervised disentanglement is fundamentally non-identifiable without suitable inductive biases or structural assumptions about the data-generating process (Esmaeili et al., 2019; Locatello et al., 2019; Schott et al., 2022). Within the $\beta$-VAE framework, performance is highly sensitive to the regularization parameter $\beta$: increasing the weight of the KL divergence restricts the informational capacity of the latent channel and can promote a more factorized aggregated posterior at the expense of reconstruction fidelity and latent informativeness. Recent high-dimensional analyses (Ichikawa & Hukushima, 2024; 2025) identify phase-transition regimes in which strong regularization induces posterior collapse, where the approximate posterior approaches the prior and becomes uninformative about the observations. Prior work has related this failure mode to properties of the ELBO objective (Lucas et al., 2019), competition between likelihood and prior regularization (Wang & Ziyin, 2022), and lagging dynamics of the inference network (He et al., 2019). However, a precise characterization of how the optimization dynamics and encoder parameterization jointly lead to a systematic loss of factor-relevant latent information remains limited.

**Robustness of Disentanglement Evaluation.** Disentangled representations are commonly evaluated using metrics such as SAP (Kumar et al., 2018), MIG (Chen et al., 2018), and DCI (Eastwood & Williams, 2018), which quantify relationships between latent variables and ground-truth generative factors through predictability, mutual information, or feature-importance measures. These metrics implicitly require the latent representation to retain sufficient information about the underlying factors to provide a meaningful evaluation signal. Although prior work has examined these properties from an information-theoretic perspective (Burgess et al., 2018), their behavior in low-informativeness regimes remains poorly understood. In particular, as latent–factor mutual information vanishes, these metrics lose discriminative power and can no longer reliably distinguish disentangled representations from uninformative ones.

**Information Retention in Generative Models.** VAEs serve as foundational representation backbones in generative pipelines, such as latent diffusion models (Rombach et al., 2022; Pynadath et al., 2025), where latent quality is a primary determinant of downstream performance. Empirical studies across diverse domains—including protein modeling (Eguchi et al., 2022), robotics (Abdulsamad et al., 2024), 3D shape representation (Jiao et al., 2024), and drug discovery (Cropsal & Mercado, 2025)—consistently report performance degradation under strong regularization, a phenomenon frequently attributed to reduced latent informativeness. To address these instabilities, recent variants such as the $\lambda\beta$-VAE and $\gamma\beta$-VAE (Vu, 2025) introduce auxiliary structural objectives. The $\lambda\beta$-VAE incorporates an $L_2$ reconstruction penalty to strengthen the reconstruction signal in high-$\beta$ regimes without compromising disentanglement. Conversely, the $\gamma\beta$-VAE utilizes a mutual information penalty within re-encoding cycles to reinforce factorization at lower $\beta$ values while maintaining high reconstruction fidelity.

## 3 Linear-Gaussian Framework

We leverage the tractability of the linear-Gaussian regime to formalize the informational collapse hypothesized in Section 1. This framework allows for a closed-form characterization of the stationarity conditions and the resulting optimization dynamics.

Figure 1: The linear-Gaussian generative and inference process. Generative factors $\mathbf{v}$ map to observations $\mathbf{y}$ via $\mathbf{\Gamma}$. The encoder $(\mathbf{B}, \mathbf{\Sigma_w})$ transforms $\mathbf{y} \to \mathbf{x}$, while the decoder $(\mathbf{A}, \mathbf{\Sigma_z})$ reconstructs $\hat{\mathbf{y}}$ from $\mathbf{x}$.

### 3.1 Generative Model

We model the informational flow within the VAE as a Markov chain $\mathbf{v} \to \mathbf{y} \to \mathbf{x} \to \hat{\mathbf{y}}$ (Figure 1). Let $\mathbf{v} \in \mathbb{R}^s$ denote the ground-truth generative factors, distributed as $\mathbf{v} \sim \mathcal{N}(\mathbf{0}, \mathbf{\Sigma_v})$, where $\mathbf{\Sigma_v} \in \mathbb{S}_{++}^s$ is diagonal and positive definite. Observations $\mathbf{y} \in \mathbb{R}^n$ are generated via a linear mixing matrix $\mathbf{\Gamma} \in \mathbb{R}^{n \times s}$ with additive Gaussian noise:

$$\mathbf{y} = \mathbf{\Gamma v} + \tilde{\mathbf{z}}, \qquad \tilde{\mathbf{z}} \sim \mathcal{N}(\mathbf{0}, \sigma^2 \mathbf{I}_n), \tag{1}$$

yielding the observation covariance

$$\mathbf{\Sigma_y} = \mathbf{\Gamma \Sigma_v \Gamma}^\top + \sigma^2 \mathbf{I}_n.$$

The latent representation $\mathbf{x} \in \mathbb{R}^m$ is subject to an isotropic Gaussian prior $p(\mathbf{x}) = \mathcal{N}(\mathbf{0}, \mathbf{I}_m)$. We define a Gaussian encoder $q_{\boldsymbol{\phi}}(\mathbf{x}|\mathbf{y})$ parameterized by the encoder gain $\mathbf{B} \in \mathbb{R}^{m \times n}$ and noise covariance $\mathbf{\Sigma_w} \in \mathbb{S}_{++}^m$:

$$\mathbf{x} = \mathbf{By} + \mathbf{w}, \qquad \mathbf{w} \sim \mathcal{N}(\mathbf{0}, \mathbf{\Sigma_w}). \tag{2}$$

The decoder $p_{\boldsymbol{\theta}}(\mathbf{y}|\mathbf{x})$ reconstructs the observations via the decoder gain $\mathbf{A} \in \mathbb{R}^{n \times m}$ and noise covariance $\mathbf{\Sigma_z} \in \mathbb{S}_{++}^n$:

$$\hat{\mathbf{y}} = \mathbf{Ax} + \mathbf{z}, \qquad \mathbf{z} \sim \mathcal{N}(\mathbf{0}, \mathbf{\Sigma_z}). \tag{3}$$

The statistical relationship between the latent variables and generative factors is captured by the joint covariance:

$$\mathbf{\Sigma} = \mathrm{Cov}\left( \begin{bmatrix} \mathbf{x} \\ \mathbf{v} \end{bmatrix} \right) = \begin{bmatrix} \mathbf{\Sigma_x} & \mathbf{\Sigma_{xv}} \\ \mathbf{\Sigma_{vx}} & \mathbf{\Sigma_v} \end{bmatrix} = \begin{bmatrix} \mathbf{B\Sigma_y B}^\top + \mathbf{\Sigma_w} & \mathbf{B\Gamma\Sigma_v} \\ \mathbf{\Sigma_v \Gamma}^\top \mathbf{B}^\top & \mathbf{\Sigma_v} \end{bmatrix}. \tag{4}$$

### 3.2 Variational Objectives

The standard $\beta$-VAE objective is formulated as a weighted ELBO, which we minimize with respect to the encoder and decoder parameters:

$$\mathcal{L}_\beta(\boldsymbol{\phi}, \boldsymbol{\theta}) = \underbrace{\mathbb{E}_{q_{\boldsymbol{\phi}}(\mathbf{x}|\mathbf{y})}\big[ -\log p_{\boldsymbol{\theta}}(\mathbf{y}|\mathbf{x}) \big]}_{\text{Reconstruction}} + \beta \underbrace{\mathrm{D_{KL}}\big( q_{\boldsymbol{\phi}}(\mathbf{x}|\mathbf{y}) \,\|\, p(\mathbf{x}) \big)}_{\text{Regularization}}. \tag{5}$$

In the linear-Gaussian regime, this objective admits a closed-form analytical expansion:

$$\mathcal{L}_\beta(\mathbf{A}, \mathbf{B}, \mathbf{\Sigma_z}, \mathbf{\Sigma_w}) = -\frac{1}{2}\Bigg( \mathrm{Tr}\Big[ \mathbf{A}^\top \mathbf{\Sigma_z}^{-1} \mathbf{\Sigma_y} \mathbf{B}^\top + \mathbf{\Sigma_z}^{-1} \mathbf{AB\Sigma_y} - \mathbf{\Sigma_z}^{-1} \mathbf{\Sigma_y}$$

$$- \mathbf{A}^\top \mathbf{\Sigma_z}^{-1} \mathbf{A}(\mathbf{B\Sigma_y B}^\top + \mathbf{\Sigma_w}) \Big] - n \log(2\pi) - \log |\mathbf{\Sigma_z}| \Bigg)$$

$$+ \frac{\beta}{2}\Big[ \mathrm{Tr}(\mathbf{B\Sigma_y B}^\top + \mathbf{\Sigma_w}) - \log |\mathbf{\Sigma_w}| - m \Big]. \tag{6}$$

As demonstrated in Section 3.3, for $\beta > 1$, the alternating-optimization dynamics induce a spectral contraction of the encoder gain, driving $\mathbf{B} \to \mathbf{0}$. This informational collapse renders the latent channel uninformative, causing evaluation metrics to lose discriminative power because they can no longer distinguish

structured representations from uninformative noise. To mitigate this failure mode, we investigate the $\lambda\beta$-VAE formulation, which augments the objective with an auxiliary $L_2$ reconstruction penalty weighted by $\lambda$. The primary goal of this term is to enhance reconstruction accuracy in high-$\beta$ regimes while retaining the latent informativeness required for meaningful disentanglement. Within the linear-Gaussian framework, we show that this intervention modifies the system's stationarity conditions and weakens the derived upper bound on encoder-gain contraction relative to the standard $\beta$-VAE. Specifically, we define the auxiliary reconstruction using $\hat{\mathbf{y}} = \mathbf{A}\mathbf{x}$, omitting the additive decoder noise $\mathbf{z}$. This approach follows prior work suggesting that deterministic decoder can reduce blurriness and improve reconstruction fidelity (Ghosh et al., 2020; Bredell et al., 2023; Kim & Lee, 2025). The auxiliary term penalizes the squared error between the reconstructed output and the observation $\mathbf{y}$:

$$\mathcal{L}_{\lambda\beta}(\boldsymbol{\phi}, \boldsymbol{\theta}) = \mathcal{L}_{\beta}(\boldsymbol{\phi}, \boldsymbol{\theta}) + \lambda \, \mathbb{E}_{q_{\boldsymbol{\phi}}(\mathbf{x}, \mathbf{y})}\big[\|\mathbf{y} - \hat{\mathbf{y}}\|^2\big], \qquad \lambda \geq 0, \tag{7}$$

where the expectation is taken over the joint distribution $q_{\boldsymbol{\phi}}(\mathbf{x}, \mathbf{y}) = q_{\boldsymbol{\phi}}(\mathbf{x}|\mathbf{y})p(\mathbf{y})$. Substituting the encoder relation $\mathbf{x} = \mathbf{B}\mathbf{y} + \mathbf{w}$ into the auxiliary term yields the augmented linear-Gaussian objective:

$$\mathcal{L}_{\lambda\beta}(\mathbf{A}, \mathbf{B}, \boldsymbol{\Sigma}_{\mathbf{z}}, \boldsymbol{\Sigma}_{\mathbf{w}}) = \mathcal{L}_{\beta}(\mathbf{A}, \mathbf{B}, \boldsymbol{\Sigma}_{\mathbf{z}}, \boldsymbol{\Sigma}_{\mathbf{w}}) + \lambda \operatorname{Tr}\left[(\mathbf{I}_n - \mathbf{A}\mathbf{B})\boldsymbol{\Sigma}_{\mathbf{y}}(\mathbf{I}_n - \mathbf{A}\mathbf{B})^\top + \mathbf{A}\boldsymbol{\Sigma}_{\mathbf{w}}\mathbf{A}^\top\right]. \tag{8}$$

The full derivations for these closed-form expressions are provided in Appendix A.

### 3.3 Informational Collapse

We characterize informational collapse through its manifestation in the linear-Gaussian regime. Specifically, this phenomenon corresponds to the asymptotic vanishing of the latent signal at the objective's stationary points. These points are governed by a system of coupled stationarity conditions, as formalized in the following lemma:

**Lemma 3.1** ($\beta$-VAE Stationarity Conditions). *Any stationary point of the $\beta$-VAE objective satisfies the following system of coupled equations for the decoder $(\mathbf{A}, \boldsymbol{\Sigma}_{\mathbf{z}})$ and encoder $(\mathbf{B}, \boldsymbol{\Sigma}_{\mathbf{w}})$:*

$$\mathbf{A} = (\boldsymbol{\Sigma}_{\mathbf{y}}^{-1} + \mathbf{B}^\top \boldsymbol{\Sigma}_{\mathbf{w}}^{-1} \mathbf{B})^{-1} \mathbf{B}^\top \boldsymbol{\Sigma}_{\mathbf{w}}^{-1} \qquad \mathbf{B} = \left[\mathbf{I}_m + \mathbf{A}^\top (\boldsymbol{\Sigma}_{\mathbf{z}}^{-1}/\beta)\mathbf{A}\right]^{-1} \mathbf{A}^\top (\boldsymbol{\Sigma}_{\mathbf{z}}^{-1}/\beta)$$

$$\boldsymbol{\Sigma}_{\mathbf{z}} = (\boldsymbol{\Sigma}_{\mathbf{y}}^{-1} + \mathbf{B}^\top \boldsymbol{\Sigma}_{\mathbf{w}}^{-1} \mathbf{B})^{-1} \qquad\qquad \boldsymbol{\Sigma}_{\mathbf{w}} = \left[\mathbf{I}_m + \mathbf{A}^\top (\boldsymbol{\Sigma}_{\mathbf{z}}^{-1}/\beta)\mathbf{A}\right]^{-1}$$

*Proof.* See Appendix B.

The $\beta^{-1}$ scaling in the encoder gain update serves as the primary driver of informational suppression. Since the optimal encoder and decoder parameters are mutually dependent, we resolve these conditions via the alternating optimization procedure detailed in Algorithm 1. This iterative process can be viewed as a continuous-variable analog of the Blahut–Arimoto algorithm (Arimoto, 1972; Blahut, 1972; Yang & Mandt, 2022), solving the closed-form stationarity condition for one parameter block while holding others fixed. For $\beta > 1$, this sequence of updates induces a spectral contraction of the encoder gain, leading to the following result:

**Theorem 3.2** (Informational Collapse for $\beta > 1$). *For any regularization strength $\beta > 1$, the alternating optimization dynamics described in Lemma 3.1 and Algorithm 1 converge to the trivial fixed point:*

$$(\mathbf{A}, \mathbf{B}, \boldsymbol{\Sigma}_{\mathbf{z}}, \boldsymbol{\Sigma}_{\mathbf{w}}) = (\mathbf{0}, \mathbf{0}, \boldsymbol{\Sigma}_{\mathbf{y}}, \mathbf{I}_m).$$

*Consequently, the encoder gain $\mathbf{B}$ vanishes, the latent representation $\mathbf{x}$ becomes statistically independent of the generative factors $\mathbf{v}$, and the mutual information decays to zero:*

$$\lim_{t \to \infty} I(\mathbf{x}; \mathbf{v})^{(t)} = 0 \quad \text{for } \beta > 1.$$

*Proof.* See Appendix D.

Within the linear-Gaussian setting, Theorem 3.2 provides a mechanistic explanation for the sharp decline in disentanglement performance observed under strong regularization. In this regime, the latent channel fails to retain factor-relevant information, rendering the representation uninformative regardless of its degree of factorization.

# 4 Disentanglement Metrics and the Role of $\lambda$

## 4.1 Analytical Metric Formulations

We consider three representative metrics across our empirical evaluations to evaluate the structural integrity of the latent space: Separated Attribute Predictability (SAP) (Kumar et al., 2018), Mutual Information Gap (MIG) (Chen et al., 2018), and the Latent Informativeness Metric (LIM) (Vu, 2025). MIG is widely used in disentanglement evaluations with discrete ground-truth generative factors, where the mutual-information gap is normalized by the discrete entropy of each factor. In the continuous linear-Gaussian setting, however, replacing discrete entropy with differential entropy does not preserve the standard bounded normalization of MIG, since differential entropy may be negative and changes under rescaling. We therefore use LIM as a continuous-variable alternative in the linear-Gaussian regime rather than extending the standard discrete MIG using differential entropy. These metrics depend on the presence of factor-relevant latent information. Consequently, within the linear-Gaussian setting, as spectral contraction drives the encoder gain toward zero and latent–factor information vanishes, SAP and LIM lose their discriminative power.

**Separated Attribute Predictability (SAP).** The SAP score utilizes the squared correlation matrix $\mathbf{S} \in \mathbb{R}^{m \times s}$, where each element $S_{i,j}$ quantifies the linear predictability of generative factor $v_j$ from latent dimension $x_i$:

$$S_{i,j} = \frac{[\mathbf{B}\boldsymbol{\Gamma}\boldsymbol{\Sigma_v}]_{i,j}^2}{[\boldsymbol{\Sigma_x}]_{i,i}[\boldsymbol{\Sigma_v}]_{j,j}}. \tag{9}$$

SAP is defined as the average difference between the two largest squared correlations for each factor:

$$\text{SAP}(\mathbf{x}, \mathbf{v}) = \frac{1}{s} \sum_{j=1}^{s} \left( S_{i^{(j)},j} - S_{i'^{(j)},j} \right), \tag{10}$$

where $i^{(j)}$ and $i'^{(j)}$ denote the indices of the most and second-most predictive latent variables for factor $v_j$, respectively. As alternating optimization drives $\mathbf{B} \to \mathbf{0}$, the numerator in the correlation expression vanishes while the latent variances $[\boldsymbol{\Sigma_x}]_{i,i}$ remain at the prior noise level. Consequently, $S_{i,j} \to 0$ for all $(i, j)$, causing the SAP score to converge to zero and lose its ability to distinguish between distinct latent structures.

**Mutual Information Gap (MIG).** MIG measures the normalized difference in mutual information between the top two latent candidates for each generative factor:

$$\text{MIG}(\mathbf{x}, \mathbf{v}) = \frac{1}{s} \sum_{j=1}^{s} \frac{I(x_{i^{(j)}}; v_j) - I(x_{i'^{(j)}}; v_j)}{H(v_j)}, \tag{11}$$

where $H(v_j)$ represents the entropy of factor $v_j$. In the linear-Gaussian regime, mutual information is a monotonic function of the squared correlation:

$$I(x_i; v_j) = -\frac{1}{2} \log(1 - S_{i,j}).$$

As $\|\mathbf{B}\|_2 \to 0$ during informational collapse, $S_{i,j} \to 0$, which in turn drives $I(x_i; v_j) \to 0$, rendering the MIG score uninformative.

**Latent Informativeness Metric (LIM).** LIM quantifies the total factor-relevant information preserved within the latent space, accounting for potential statistical correlations between generative factors $\mathbf{v}$. It identifies an optimal partition $\mathcal{P} = \{v_{s_k}\}_{k=1}^{m}$ of the $s$ factors into $m$ disjoint subsets, each assigned to a latent dimension $x_k$:

$$\text{LIM}(\mathbf{x}, \mathbf{v}) = \max_{\mathcal{P}} \left[ \sum_{k=1}^{m} I(x_k; v_{s_k}) - \sum_{i<j} I(v_{s_i}; v_{s_j}) \right]. \tag{12}$$

The second term serves to remove informational redundancy among factor groups. In our controlled linear-Gaussian setting with independent factors ($\boldsymbol{\Sigma_v}$ diagonal), this redundancy term vanishes, reducing LIM to

a maximum-weight assignment problem. Furthermore, the linear-Gaussian assumption enables the exact computation of $I(x_k; v_{s_k})$ via log-determinants of covariance sub-blocks, bypassing the numerical estimation issues that MIG can exhibit for continuous variables.

## 4.2 Effect of the $\lambda$-Penalty on Latent Informativeness

In the standard $\beta$-VAE, within the linear-Gaussian setting and under alternating optimization, $\beta > 1$ induces a spectral contraction of the encoder gain, driving $\|\mathbf{B}\|_2 \to 0$ and causing latent–factor mutual information to vanish. As the latent representation becomes uninformative about the generative factors, informativeness-related metrics lose their discriminative power. To examine how an auxiliary reconstruction penalty changes this behavior, the $\lambda\beta$-VAE augments the objective with a $\lambda$-weighted $L_2$ reconstruction term, thereby modifying the encoder stationarity conditions.

**Lemma 4.1** ($\lambda\beta$-VAE Stationarity Conditions). *Let $\mathbf{M} := \beta^{-1}(\mathbf{\Sigma_z}^{-1} + 2\lambda\mathbf{I}_n)$. At any stationary point of the $\lambda\beta$-VAE objective, the decoder parameters $(\mathbf{A}, \mathbf{\Sigma_z})$ satisfy the conditions established in Lemma 3.1, while the encoder parameters satisfy the following augmented system:*

$$\mathbf{B} = (\mathbf{I}_m + \mathbf{A}^\top \mathbf{M} \mathbf{A})^{-1} \mathbf{A}^\top \mathbf{M}$$
$$\mathbf{\Sigma_w} = (\mathbf{I}_m + \mathbf{A}^\top \mathbf{M} \mathbf{A})^{-1}.$$

*Proof.* See Appendix C.

Compared with the standard $\beta$-VAE, the $\lambda$-weighted reconstruction penalty weakens the derived upper bound on encoder-gain contraction. Specifically, the condition under which the bound establishes exponential decay changes from $\beta > 1$ to $\beta > 1 + 2\lambda\|\mathbf{\Sigma_y}\|_2$. Accordingly, when $\lambda > 0$, a larger value of $\beta$ is required before the bound establishes exponential decay, so latent informativeness may be retained over a broader range of regularization strengths. This does not, however, prove that latent–factor mutual information is preserved. For fixed $\lambda$, the condition is satisfied when $\beta$ is sufficiently large, in which case the bound still establishes exponential encoder-gain decay and, therefore, vanishing latent–factor mutual information. The full derivation is provided in Appendix E.

## 5 Empirical Evaluation

We evaluate the $\lambda\beta$-VAE framework through two complementary studies: (i) controlled linear-Gaussian simulations that examine the optimization behavior analyzed theoretically in Section 3.3, and (ii) experiments with deep convolutional architectures that assess whether similar empirical trends arise in nonlinear settings.[1]

### 5.1 Linear-Gaussian Regime

We first examine the trivial-solution behavior predicted for $\beta > 1$ using the configuration $n = 50$, $m = 10$, and $s = 5$. The generative factors $\mathbf{v}$ are sampled i.i.d., with observation-noise variance $\sigma^2 = 0.05$. We compare two optimization procedures: **Alternating Optimization** (Algorithm 1), which uses the closed-form stationarity conditions established in Lemma 3.1, and **AdamW**, implemented using Cholesky-parameterized covariance matrices. For each $(\beta, \lambda)$ configuration, we perform 100 independent trials with randomized initializations $(\mathbf{B}^{(0)}, \mathbf{\Sigma_w}^{(0)})$ to assess variability across runs.

**Empirical Findings.** For the standard $\beta$-VAE, alternating optimization exhibits informational collapse at every evaluated value of $\beta > 1$ (Figure 2), consistent with Theorem 3.2. In this regime, $I(\mathbf{x}; \mathbf{v})$, SAP, and LIM converge to zero, while reconstruction error increases substantially. AdamW exhibits the same qualitative behavior over the evaluated grid, although its dynamics are not covered by the alternating-optimization theorem. Thus, informational collapse is observed empirically under both optimization procedures.

Figure 3 shows how this behavior changes when $\lambda > 0$. Relative to the $\lambda = 0$ baseline, larger values of $\lambda$ are generally associated with lower reconstruction error and higher values of $I(\mathbf{x}; \mathbf{v})$, SAP, and LIM across

---

[1]Code repository: https://github.com/mh-vu/lambda-beta-vae

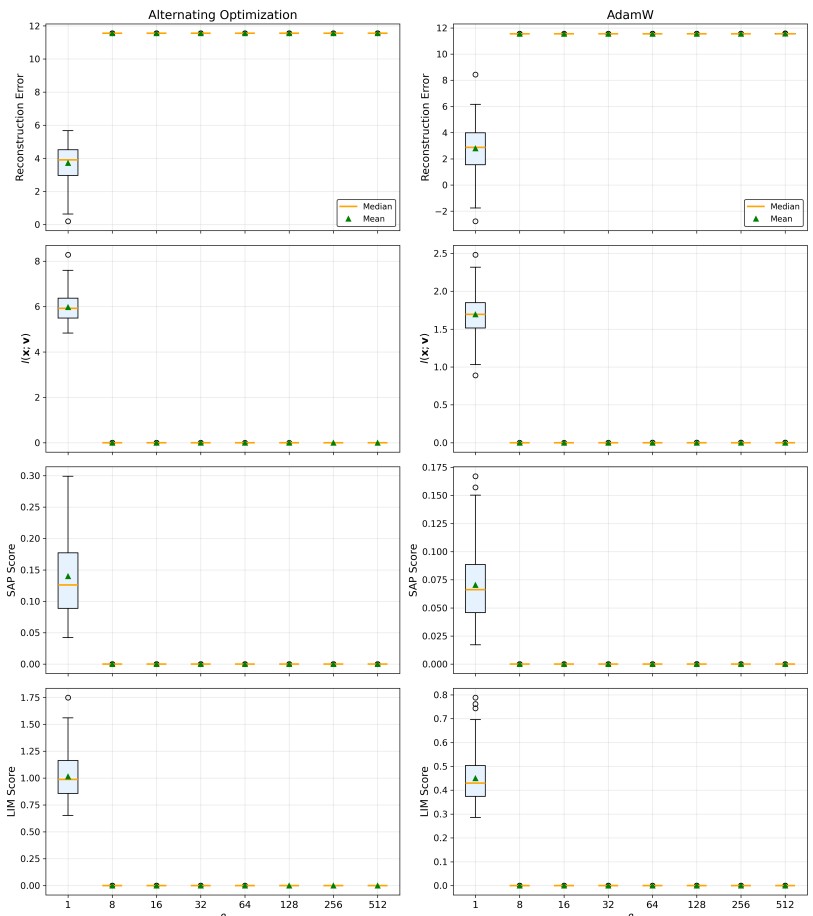

Figure 2: Distributions of reconstruction error, $I(\mathbf{x}; \mathbf{v})$, SAP score, and LIM score for the standard $\beta$-VAE ($n = 50$, $m = 10$, $s = 5$) under alternating optimization and AdamW. Each boxplot summarizes 100 independent runs with random initialization for a fixed value of $\beta$, showing the median, interquartile range, whiskers, and outliers. The mean is marked by a triangle.

a broad range of the evaluated $\beta$ values. For each fixed $\lambda > 0$, however, the informativeness-related metrics generally decline as $\beta$ increases. These results indicate that the auxiliary reconstruction penalty attenuates, but does not eliminate, the deterioration associated with strong regularization. Each curve in Figure 3 reports the mean over 100 independent random initializations for a fixed $(\beta, \lambda)$ configuration, with the shaded regions denoting 95% bootstrap confidence intervals. The intervals are generally narrow, indicating that the mean trends are consistent across random initializations. Additional heatmaps and boxplots are provided in Appendix G. The heatmaps summarize mean performance over the $(\beta, \lambda)$ grid, while the boxplots show the corresponding distributions across runs. These visualizations support the same qualitative patterns as the line plots: informational collapse occurs for the standard $\beta$-VAE at all evaluated values of $\beta > 1$, whereas configurations with $\lambda > 0$ generally achieve higher reconstruction accuracy and higher values of the informativeness-related metrics over broader ranges of $\beta$.

Overall, the linear experiments indicate that positive values of $\lambda$ are associated with higher reconstruction accuracy and attenuated deterioration in latent informativeness and, consequently, disentanglement performance relative to the standard $\beta$-VAE. This empirical pattern is qualitatively consistent with the weaker upper bound on encoder-gain contraction derived for the $\lambda\beta$-VAE. It does not, however, guarantee non-vanishing latent informativeness. For fixed $\lambda$, the derived bound remains contractive when $\beta$ is sufficiently large.

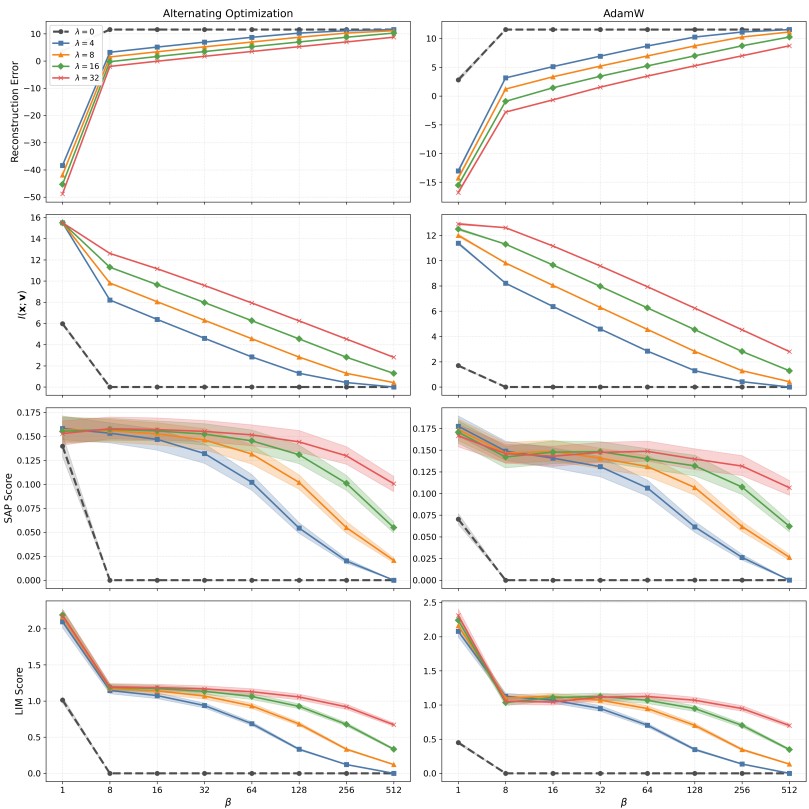

Figure 3: Mean reconstruction error, $I(\mathbf{x}; \mathbf{v})$, SAP score, and LIM score for the $\lambda\beta$-VAE ($n = 50$, $m = 10$, $s = 5$) under alternating optimization and AdamW. Each curve shows performance as a function of $\beta$ for a fixed value of $\lambda$. Points represent means over 100 independent runs with random initialization for each $(\beta, \lambda)$ configuration, and shaded regions denote 95% bootstrap confidence intervals.

## 5.2 Deep Nonlinear Architectures

To investigate whether trends similar to those observed in the linear-Gaussian analysis also arise in deep non-linear models, we evaluate the $\lambda\beta$-VAE on three standard disentanglement benchmarks: dSprites (Matthey et al., 2017), Shapes3D (Burgess & Kim, 2018), and MPI3D-real (Gondal et al., 2019). MPI3D-real, in particular, contains more realistic visual correlations and backgrounds, providing a more challenging setting than the synthetic benchmarks (Schott et al., 2022).

Table 1: Convolutional VAE architecture used in the nonlinear experiments.

| Encoder | Decoder |
|---|---|
| Input: $64 \times 64 \times C$ | Input: $\mathbf{x} \in \mathbb{R}^m$ |
| $4 \times 4$ Conv, 32, stride 2, ReLU | FC 256, ReLU |
| $4 \times 4$ Conv, 32, stride 2, ReLU | FC $4 \times 4 \times 64$, ReLU |
| $4 \times 4$ Conv, 64, stride 2, ReLU | $4 \times 4$ ConvTranspose, 64, stride 2, ReLU |
| $4 \times 4$ Conv, 64, stride 2, ReLU | $4 \times 4$ ConvTranspose, 32, stride 2, ReLU |
| FC 256, ReLU | $4 \times 4$ ConvTranspose, 32, stride 2, ReLU |
| FC $2m$ ($\boldsymbol{\mu}, \log \boldsymbol{\sigma}^2$) | $4 \times 4$ ConvTranspose, $C$, stride 2 |

**Architecture and Training Protocol.** We employ a deep convolutional VAE with a 15-dimensional latent space, a Gaussian encoder, and a Bernoulli decoder, following the architecture used by Locatello et al. (2019). Detailed architectural specifications and hyperparameter settings are provided in Tables 1 and 2.

Table 2: Hyperparameter configurations used in the nonlinear experiments.

| Hyperparameter | Value |
|---|---|
| Optimizer | Adam |
| Learning rate | $10^{-4}$ |
| Batch size | 64 |
| Latent dimension ($m$) | 15 |
| $\beta$ grid | $\{1, 4, 8, 16, 32, 64, 128, 256\}$ |
| $\lambda$ grid (Phase 2) | $\{0, 4, 8, 16, 32\}$ |
| Phase 1 pretraining steps | 150,000 |
| Phase 2 continuation steps | 50,000 |
| Total training steps per configuration | 200,000 |
| Number of seeds | 10 |

All configurations are evaluated using a unified two-stage training protocol with a total training budget of 200,000 steps:

- **Phase 1 (Pretraining):** For each value of $\beta$, a standard $\beta$-VAE is trained for 150,000 steps to produce a common initialization checkpoint for the corresponding Phase 2 configurations.
- **Phase 2 (Continued Training):** Starting from the corresponding pretrained checkpoint, training continues for an additional 50,000 steps under the $\lambda\beta$-VAE objective for each $\lambda \in \{0, 4, 8, 16, 32\}$.

The Phase 2 objective is the $\lambda\beta$-VAE objective introduced in Section 3.2:

$$\mathcal{L}_{\lambda\beta}(\boldsymbol{\phi}, \boldsymbol{\theta}) = \mathcal{L}_{\beta}(\boldsymbol{\phi}, \boldsymbol{\theta}) + \lambda \, \mathbb{E}_{q_{\boldsymbol{\phi}}(\mathbf{x}, \mathbf{y})}\big[\|\mathbf{y} - \hat{\mathbf{y}}\|^2\big], \qquad \lambda \geq 0.$$

When $\lambda = 0$, the Phase 2 objective reduces exactly to the standard $\beta$-VAE objective, and the baseline is continued for the full Phase 2 budget rather than being evaluated only at the end of pretraining. This design provides a matched-budget comparison in which, for each fixed $\beta$, the Phase 2 objectives differ only in the value of $\lambda$.

The theoretical results in Sections 3.3–4.2 apply only to the linear-Gaussian setting under the specified alternating-optimization dynamics. The nonlinear experiments instead examine whether related empirical trends—including changes in reconstruction error and disentanglement performance as $\beta$ and $\lambda$ vary—also arise in deep convolutional architectures. These experiments are not intended to validate or extend the theoretical results beyond the assumptions of the linear-Gaussian model.

**Empirical Findings.** Figure 4 shows the behavior of the standard $\beta$-VAE baseline across the three datasets. As $\beta$ increases, reconstruction error generally increases, whereas MIG and SAP vary non-monotonically. Both metrics typically improve at low or moderate regularization strengths before declining as $\beta$ becomes large. Across all three datasets, MIG and SAP decrease from their peak values in the strongly regularized regime. On MPI3D-real, this decline begins at smaller values of $\beta$ and persists as the regularization strength increases, consistent with prior observations that more complex visual datasets may be more sensitive to strong regularization (Ichikawa & Hukushima, 2025).

Figure 5 extends this analysis to the $\lambda\beta$-VAE. Across all three datasets, configurations with $\lambda > 0$ generally achieve lower reconstruction error than the $\lambda = 0$ baseline over most of the evaluated $\beta$ range. The effects on MIG and SAP are less consistent at low and moderate values of $\beta$, where the curves often overlap and no single value of $\lambda$ consistently performs best. At larger values of $\beta$, corresponding to the strongly regularized regime, configurations with $\lambda > 0$ generally attain higher MIG and SAP scores than the $\lambda = 0$ baseline. Additional heatmaps and boxplots are provided in Appendix G. The heatmaps summarize mean performance over the $(\beta, \lambda)$ grid, while the boxplots show the corresponding distributions across runs. These complementary visualizations support the qualitative trends observed in Figure 5 and further illustrate that variability across seeds depends on the dataset, metric, and hyperparameter configuration.

Overall, the nonlinear experiments indicate that positive values of $\lambda$ are generally associated with increased reconstruction accuracy and an attenuated decline in disentanglement performance over the evaluated range

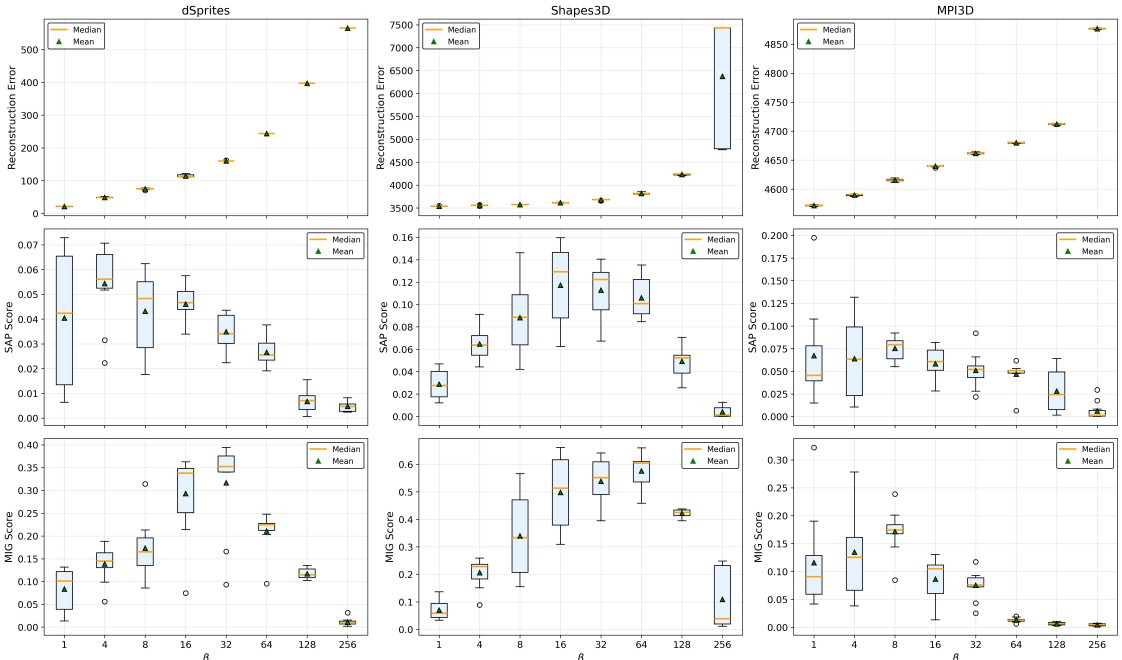

Figure 4: Distributions of reconstruction error, SAP score, and MIG score for the standard $\beta$-VAE baseline on dSprites, Shapes3D, and MPI3D-real. Each boxplot summarizes 10 independent training runs with different random seeds for a fixed value of $\beta$, showing the median, interquartile range, whiskers, and outliers. The mean is marked by a triangle.

of $\beta$. These effects are most evident at large values of $\beta$, where the standard $\beta$-VAE often exhibits substantial degradation in both MIG and SAP. Although the overall trend is consistent across the evaluated datasets, no single value of $\lambda$ performs best across every dataset and metric. The nonlinear results therefore provide complementary empirical evidence from deep convolutional architectures rather than a validation or extension of the linear-Gaussian theory.

## 6 Hyperparameter Trade-off Analysis

To characterize the trade-off between reconstruction fidelity and disentanglement, we perform a multi-objective benchmark analysis over the evaluated $(\beta, \lambda)$ grid using ground-truth generative factors. This controlled analysis examines how different configurations balance the two objectives under benchmark conditions where such factors are available. We define the objective vector $\mathbf{f} = [f_1, f_2]^\top$, where $f_1$ denotes reconstruction error, quantified by the negative log-likelihood, and $f_2 = 1 - \mathrm{MIG}$ denotes the *entanglement cost*. To navigate the trade-off between these objectives, we use **augmented Tchebycheff scalarization** (Steuer & Choo, 1983; Miettinen, 1999; Dächert et al., 2012; Chugh, 2020). Each objective is first normalized over the evaluation grid $\mathcal{G}$ to ensure scale invariance:

$$\bar{f}_i(\beta, \lambda) = \frac{f_i(\beta, \lambda) - \min_{\mathcal{G}} f_i}{\max_{\mathcal{G}} f_i - \min_{\mathcal{G}} f_i}, \quad i \in \{1, 2\}. \tag{13}$$

Given a preference vector $\boldsymbol{\omega}$ satisfying $\omega_1 + \omega_2 = 1$, we seek to minimize the scalarized selection score $\mathcal{S}$:

$$\mathcal{S}(\beta, \lambda; \boldsymbol{\omega}) = \max_{i \in \{1, 2\}} \{\omega_i \bar{f}_i(\beta, \lambda)\} + \rho \sum_{i=1}^{2} \omega_i \bar{f}_i(\beta, \lambda), \tag{14}$$

where $\rho = 10^{-3}$ is a small augmentation parameter that promotes strict Pareto optimality.

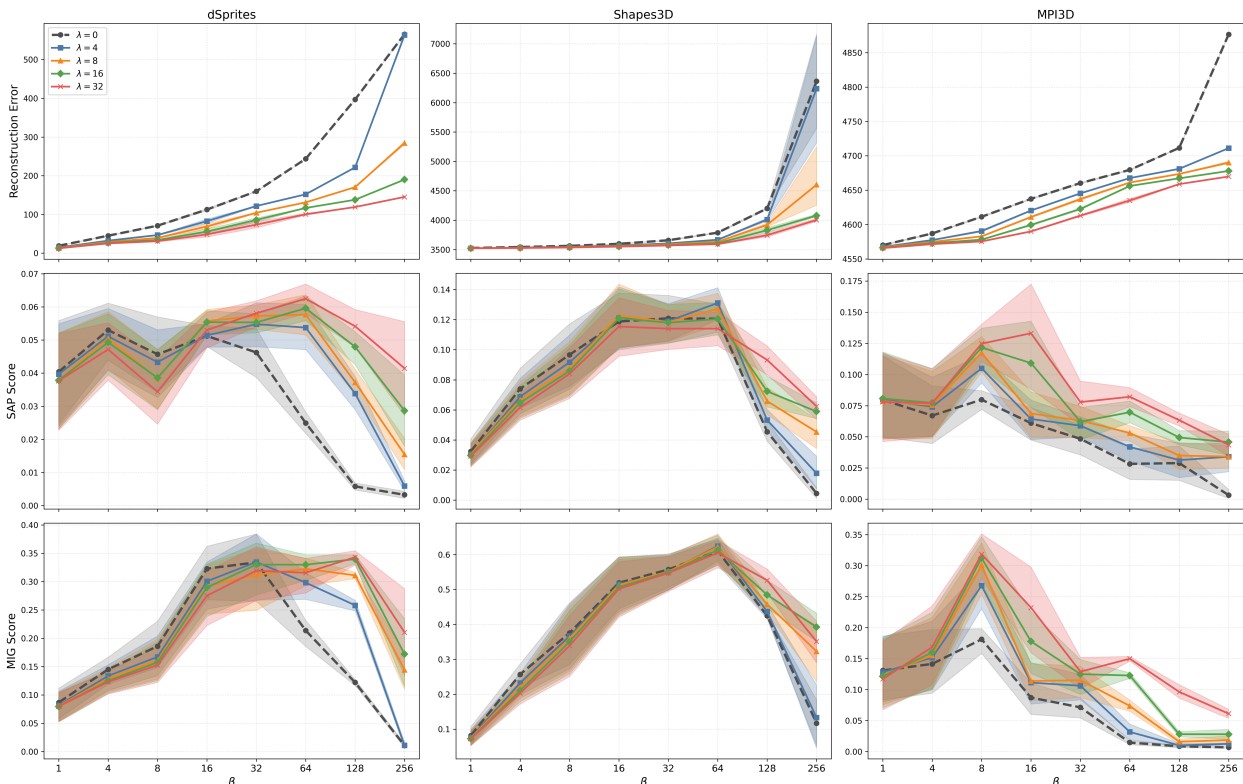

Figure 5: Mean reconstruction error, SAP score, and MIG score for the $\lambda\beta$-VAE on dSprites, Shapes3D, and MPI3D-real. Each curve shows performance as a function of $\beta$ for a fixed value of $\lambda$. Points represent means over 10 independent training runs with different random seeds for each $(\beta, \lambda)$ configuration, and shaded regions denote 95% bootstrap confidence intervals.

**Empirical Findings.** Figure 6 shows the scalarization scores under three representative preference profiles. As the priority shifts from reconstruction to disentanglement, the selected configurations tend to move from lower-$\beta$ regions toward stronger regularization. Within strongly regularized regions, configurations with $\lambda > 0$ often achieve lower reconstruction error while maintaining competitive MIG values relative to the corresponding $\lambda = 0$ configurations. The preferred $(\beta, \lambda)$ pair nevertheless depends on the dataset and the chosen preference profile, indicating that no single configuration universally optimizes the reconstruction–disentanglement trade-off. These results provide a retrospective analysis of the trade-offs observed across the tested configurations under benchmark conditions where ground-truth generative factors are available. Because MIG requires access to these factors, this analysis is not a deployable hyperparameter-selection strategy and cannot be applied directly in general unsupervised settings. Developing practical criteria for hyperparameter selection in fully unsupervised settings remains an important direction for future work.

## 7 Conclusion

This work investigates informational collapse as a failure mode in strongly regularized $\beta$-VAEs. We interpret the commonly observed non-monotonic behavior of disentanglement performance through the loss of factor-related information in the latent representation. Within a linear-Gaussian framework and under the specified alternating-optimization procedure, we prove that, for $\beta > 1$, the encoder gain undergoes spectral contraction, causing its spectral norm to decay exponentially and latent–factor mutual information to vanish. Within this setting, the result provides a mechanistic explanation for the loss of latent informativeness, which is accompanied by deteriorating disentanglement performance in the corresponding experiments.

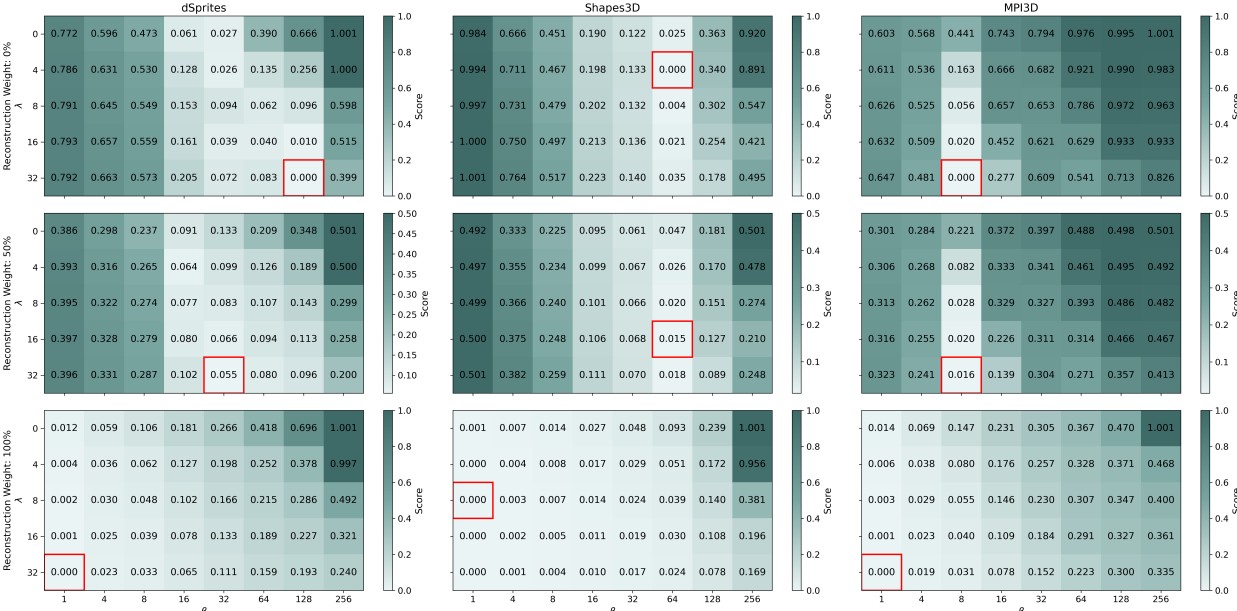

Figure 6: Augmented Tchebycheff scalarization scores under three representative preference profiles. Red markers indicate the $(\beta, \lambda)$ configurations that minimize the scalarized score over the evaluated grid $\mathcal{G}$.

Motivated by this analysis, we investigate the $\lambda\beta$-VAE, which augments the objective with an auxiliary $L_2$ reconstruction penalty. In the linear-Gaussian setting, $\lambda$ modifies the encoder stationarity conditions and weakens the derived upper bound on encoder-gain contraction relative to the standard $\beta$-VAE. This analysis does not guarantee non-vanishing latent–factor mutual information; for fixed $\lambda$, the derived bound remains contractive when $\beta$ is sufficiently large. Empirically, positive values of $\lambda$ are generally associated with increased reconstruction accuracy and attenuated deterioration of latent informativeness and disentanglement performance over the evaluated ranges of $\beta$, particularly under strong regularization. No single value of $\lambda$ consistently performs best across all datasets and hyperparameter configurations. The nonlinear findings should therefore be interpreted as empirical observations rather than as a validation or extension of the linear-Gaussian theorem.

Retaining factor-related information does not resolve the inherent non-identifiability of unsupervised disentanglement (Locatello et al., 2019). Rather, it maintains the factor-relevant signal that underlies disentanglement metrics and can support related downstream analyses. These findings motivate several promising directions for future research, including developing practical criteria for dynamically tuning $\lambda$ without relying on ground-truth generative factors, potentially through control-theoretic approaches (Shao et al., 2020); investigating latent informativeness in modern generative pipelines, such as latent diffusion models (Rombach et al., 2022; Pynadath et al., 2025); and extending the analysis to high-dimensional asymptotics (Ichikawa & Hukushima, 2024; 2025; Rahimi et al., 2025) and alternative latent geometries (Cho et al., 2023; Pynadath et al., 2025).

Overall, our results suggest a design principle within the settings studied: strong regularization can erode the factor-related information required for meaningful disentanglement, whereas an auxiliary reconstruction penalty may attenuate this deterioration, particularly in strongly regularized regimes.

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

# A  Analytical Expansion of the $\beta$-VAE and $\lambda\beta$-VAE

## A.1  Regularization Term

*Proof.* The regularization term in the $\beta$-VAE objective is defined as the expected KL divergence between the approximate posterior $q_{\phi}(\mathbf{x}|\mathbf{y})$ and the prior $p(\mathbf{x})$. In the linear-Gaussian regime, this expectation over the data distribution $p(\mathbf{y})$ is equivalent to the KL divergence between the joint distribution and the product of marginals:

$$\mathbb{E}_{p(\mathbf{y})}\left[\mathrm{D}_{\mathrm{KL}}(q_{\phi}(\mathbf{x}|\mathbf{y})\|p(\mathbf{x}))\right] = \mathrm{D}_{\mathrm{KL}}(q_{\phi}(\mathbf{x},\mathbf{y})\|p(\mathbf{x})p(\mathbf{y})),$$

where $q_{\phi}(\mathbf{x},\mathbf{y}) = q_{\phi}(\mathbf{x}|\mathbf{y})p(\mathbf{y})$. Under the encoder mapping $\mathbf{x} = \mathbf{B}\mathbf{y} + \mathbf{w}$, the joint distribution $q_{\phi}(\mathbf{x},\mathbf{y})$ and the product of marginals $p(\mathbf{x})p(\mathbf{y})$ are zero-mean multivariate Gaussians:

$$q_{\phi}(\mathbf{x},\mathbf{y}) \sim \mathcal{N}(\mathbf{0},\boldsymbol{\Sigma}_q), \quad p(\mathbf{x})p(\mathbf{y}) \sim \mathcal{N}(\mathbf{0},\boldsymbol{\Sigma}_p),$$

with the respective joint covariance matrices defined as:

$$\boldsymbol{\Sigma}_q = \begin{bmatrix} \mathbf{B}\boldsymbol{\Sigma}_{\mathbf{y}}\mathbf{B}^{\top} + \boldsymbol{\Sigma}_{\mathbf{w}} & \mathbf{B}\boldsymbol{\Sigma}_{\mathbf{y}} \\ \boldsymbol{\Sigma}_{\mathbf{y}}\mathbf{B}^{\top} & \boldsymbol{\Sigma}_{\mathbf{y}} \end{bmatrix}, \quad \boldsymbol{\Sigma}_p = \begin{bmatrix} \mathbf{I}_m & \mathbf{0} \\ \mathbf{0} & \boldsymbol{\Sigma}_{\mathbf{y}} \end{bmatrix}.$$

Thus, the KL divergence between these two distributions is given by:

$$
\begin{aligned}
D_{\mathrm{KL}} &= \frac{1}{2}\left[\log\frac{|\boldsymbol{\Sigma}_p|}{|\boldsymbol{\Sigma}_q|} - (m+n) + \mathrm{Tr}\left(\boldsymbol{\Sigma}_p^{-1}\boldsymbol{\Sigma}_q\right)\right] \\
&= \frac{1}{2}\left[\log\frac{|\mathbf{I}_m||\boldsymbol{\Sigma}_{\mathbf{y}}|}{|\boldsymbol{\Sigma}_{\mathbf{w}}||\boldsymbol{\Sigma}_{\mathbf{y}}|} - (m+n) + \mathrm{Tr}\left(\begin{bmatrix}\mathbf{I}_m & \mathbf{0} \\ \mathbf{0} & \boldsymbol{\Sigma}_{\mathbf{y}}^{-1}\end{bmatrix}\begin{bmatrix}\mathbf{B}\boldsymbol{\Sigma}_{\mathbf{y}}\mathbf{B}^\top + \boldsymbol{\Sigma}_{\mathbf{w}} & \mathbf{B}\boldsymbol{\Sigma}_{\mathbf{y}} \\ \boldsymbol{\Sigma}_{\mathbf{y}}\mathbf{B}^\top & \boldsymbol{\Sigma}_{\mathbf{y}}\end{bmatrix}\right)\right] \\
&= \frac{1}{2}\left[-\log|\boldsymbol{\Sigma}_{\mathbf{w}}| - m - n + \mathrm{Tr}\begin{bmatrix}\mathbf{B}\boldsymbol{\Sigma}_{\mathbf{y}}\mathbf{B}^\top + \boldsymbol{\Sigma}_{\mathbf{w}} & \mathbf{B}\boldsymbol{\Sigma}_{\mathbf{y}} \\ \mathbf{B}^\top & \mathbf{I}_n\end{bmatrix}\right] \\
&= \frac{1}{2}\left[-\log|\boldsymbol{\Sigma}_{\mathbf{w}}| - m - n + \mathrm{Tr}(\mathbf{B}\boldsymbol{\Sigma}_{\mathbf{y}}\mathbf{B}^\top + \boldsymbol{\Sigma}_{\mathbf{w}}) + n\right] \\
&= \frac{1}{2}\left[\mathrm{Tr}(\mathbf{B}\boldsymbol{\Sigma}_{\mathbf{y}}\mathbf{B}^\top + \boldsymbol{\Sigma}_{\mathbf{w}}) - \log|\boldsymbol{\Sigma}_{\mathbf{w}}| - m\right].
\end{aligned}
$$

$\square$

## A.2 Reconstruction Term

*Proof.* The reconstruction term $\mathcal{L}_{\mathrm{recon}}$ is defined as the expected negative log-likelihood of the decoder. We can rewrite this expectation over the joint distribution $q_{\boldsymbol{\phi}}(\mathbf{x},\mathbf{y})$:

$$\mathcal{L}_{\mathrm{recon}} = -\mathbb{E}_{q_{\boldsymbol{\phi}}(\mathbf{x},\mathbf{y})}[\log p_{\boldsymbol{\theta}}(\mathbf{y}|\mathbf{x})].$$

Utilizing the conditional probability identity $\log p_{\boldsymbol{\theta}}(\mathbf{y}|\mathbf{x}) = \log p_{\boldsymbol{\theta}}(\mathbf{x},\mathbf{y}) - \log p(\mathbf{x})$, we can decompose the reconstruction term as:

$$\mathcal{L}_{\mathrm{recon}} = \underbrace{-\mathbb{E}_{q_{\boldsymbol{\phi}}(\mathbf{x},\mathbf{y})}\left[\log p_{\boldsymbol{\theta}}(\mathbf{x},\mathbf{y})\right]}_{\text{Joint Term}} + \underbrace{\mathbb{E}_{q_{\boldsymbol{\phi}}(\mathbf{x},\mathbf{y})}\left[\log p(\mathbf{x})\right]}_{\text{Prior Term}}. \tag{15}$$

Using the decoder mapping $\hat{\mathbf{y}} = \mathbf{A}\mathbf{x} + \mathbf{z}$, the joint distribution $p_{\boldsymbol{\theta}}(\mathbf{x},\mathbf{y})$ is given by $\mathcal{N}(\mathbf{0}, \boldsymbol{\Sigma}_{\boldsymbol{\theta}})$, where the covariance and its inverse are:

$$\boldsymbol{\Sigma}_{\boldsymbol{\theta}} = \begin{bmatrix}\mathbf{I}_m & \mathbf{A}^\top \\ \mathbf{A} & \mathbf{A}\mathbf{A}^\top + \boldsymbol{\Sigma}_{\mathbf{z}}\end{bmatrix}, \quad \boldsymbol{\Sigma}_{\boldsymbol{\theta}}^{-1} = \begin{bmatrix}\mathbf{I}_m + \mathbf{A}^\top\boldsymbol{\Sigma}_{\mathbf{z}}^{-1}\mathbf{A} & -\mathbf{A}^\top\boldsymbol{\Sigma}_{\mathbf{z}}^{-1} \\ -\boldsymbol{\Sigma}_{\mathbf{z}}^{-1}\mathbf{A} & \boldsymbol{\Sigma}_{\mathbf{z}}^{-1}\end{bmatrix}.$$

The joint term in (15) is expressed using the multivariate Gaussian log-likelihood as follows:

$$
\begin{aligned}
&-\mathbb{E}_{q_{\boldsymbol{\phi}}(\mathbf{x},\mathbf{y})}\left[\log p_{\boldsymbol{\theta}}(\mathbf{x},\mathbf{y})\right] \\
&= \frac{m+n}{2}\log(2\pi) + \frac{1}{2}\log|\boldsymbol{\Sigma}_{\boldsymbol{\theta}}| + \frac{1}{2}\mathbb{E}_{q_{\boldsymbol{\phi}}(\mathbf{x},\mathbf{y})}\left[\begin{bmatrix}\mathbf{x} \\ \mathbf{y}\end{bmatrix}^\top \boldsymbol{\Sigma}_{\boldsymbol{\theta}}^{-1}\begin{bmatrix}\mathbf{x} \\ \mathbf{y}\end{bmatrix}\right] \\
&= \frac{m+n}{2}\log(2\pi) + \frac{1}{2}\log|\boldsymbol{\Sigma}_{\boldsymbol{\theta}}| + \frac{1}{2}\mathrm{Tr}\left(\boldsymbol{\Sigma}_{\boldsymbol{\theta}}^{-1}\mathbb{E}_{q_{\boldsymbol{\phi}}(\mathbf{x},\mathbf{y})}\left[\begin{bmatrix}\mathbf{x} \\ \mathbf{y}\end{bmatrix}\begin{bmatrix}\mathbf{x} \\ \mathbf{y}\end{bmatrix}^\top\right]\right) \\
&= \frac{m+n}{2}\log(2\pi) + \frac{1}{2}\log|\boldsymbol{\Sigma}_{\boldsymbol{\theta}}| + \frac{1}{2}\mathrm{Tr}\left(\boldsymbol{\Sigma}_{\boldsymbol{\theta}}^{-1}\boldsymbol{\Sigma}_q\right) \\
&= \frac{m+n}{2}\log(2\pi) + \frac{1}{2}\log|\boldsymbol{\Sigma}_{\mathbf{z}}| + \frac{1}{2}\mathrm{Tr}\left(\begin{bmatrix}\mathbf{I}_m + \mathbf{A}^\top\boldsymbol{\Sigma}_{\mathbf{z}}^{-1}\mathbf{A} & -\mathbf{A}^\top\boldsymbol{\Sigma}_{\mathbf{z}}^{-1} \\ -\boldsymbol{\Sigma}_{\mathbf{z}}^{-1}\mathbf{A} & \boldsymbol{\Sigma}_{\mathbf{z}}^{-1}\end{bmatrix}\begin{bmatrix}\boldsymbol{\Sigma}_{\mathbf{x}} & \mathbf{B}\boldsymbol{\Sigma}_{\mathbf{y}} \\ \boldsymbol{\Sigma}_{\mathbf{y}}\mathbf{B}^\top & \boldsymbol{\Sigma}_{\mathbf{y}}\end{bmatrix}\right) \\
&= \frac{m+n}{2}\log(2\pi) + \frac{1}{2}\log|\boldsymbol{\Sigma}_{\mathbf{z}}| + \frac{1}{2}\mathrm{Tr}\left(\boldsymbol{\Sigma}_{\mathbf{x}} + \mathbf{A}^\top\boldsymbol{\Sigma}_{\mathbf{z}}^{-1}\mathbf{A}\boldsymbol{\Sigma}_{\mathbf{x}} - \mathbf{A}^\top\boldsymbol{\Sigma}_{\mathbf{z}}^{-1}\boldsymbol{\Sigma}_{\mathbf{y}}\mathbf{B}^\top - \boldsymbol{\Sigma}_{\mathbf{z}}^{-1}\mathbf{A}\mathbf{B}\boldsymbol{\Sigma}_{\mathbf{y}} + \boldsymbol{\Sigma}_{\mathbf{z}}^{-1}\boldsymbol{\Sigma}_{\mathbf{y}}\right).
\end{aligned}
$$

With the isotropic Gaussian prior $p(\mathbf{x}) = \mathcal{N}(\mathbf{0}, \mathbf{I}_m)$, the prior term in (15) is:

$$\mathbb{E}_{q_{\boldsymbol{\phi}}(\mathbf{x},\mathbf{y})}\left[\log p(\mathbf{x})\right] = \mathbb{E}_{q_{\boldsymbol{\phi}}(\mathbf{x},\mathbf{y})}\left[-\frac{m}{2}\log(2\pi) - \frac{1}{2}\mathbf{x}^\top\mathbf{x}\right] = -\frac{m}{2}\log(2\pi) - \frac{1}{2}\mathrm{Tr}(\boldsymbol{\Sigma}_{\mathbf{x}}).$$

Combining the joint and prior terms yields the final reconstruction term:

$$\mathcal{L}_{\mathrm{recon}} = \frac{n}{2}\log(2\pi) + \frac{1}{2}\log|\boldsymbol{\Sigma}_{\mathbf{z}}| + \frac{1}{2}\mathrm{Tr}\left(\mathbf{A}^\top\boldsymbol{\Sigma}_{\mathbf{z}}^{-1}\mathbf{A}\boldsymbol{\Sigma}_{\mathbf{x}} - \mathbf{A}^\top\boldsymbol{\Sigma}_{\mathbf{z}}^{-1}\boldsymbol{\Sigma}_{\mathbf{y}}\mathbf{B}^\top - \boldsymbol{\Sigma}_{\mathbf{z}}^{-1}\mathbf{A}\mathbf{B}\boldsymbol{\Sigma}_{\mathbf{y}} + \boldsymbol{\Sigma}_{\mathbf{z}}^{-1}\boldsymbol{\Sigma}_{\mathbf{y}}\right),$$

where $\boldsymbol{\Sigma}_{\mathbf{x}} = \mathbf{B}\boldsymbol{\Sigma}_{\mathbf{y}}\mathbf{B}^\top + \boldsymbol{\Sigma}_{\mathbf{w}}$.

$\square$

### A.3 $L_2$ Reconstruction Penalty

*Proof.* Under the encoder mapping $\mathbf{x} = \mathbf{B}\mathbf{y} + \mathbf{w}$, the $L_2$ reconstruction penalty is evaluated by substituting the inference relation directly into the quadratic form:

$$\mathbb{E}_{q_\phi(\mathbf{x},\mathbf{y})}\big[\|\mathbf{y} - \hat{\mathbf{y}}\|^2\big]$$

$$
\begin{aligned}
&= \mathbb{E}_{q_\phi(\mathbf{x},\mathbf{y})}[\|\mathbf{y} - \mathbf{A}\mathbf{x}\|^2] \\
&= \mathbb{E}_{q_\phi(\mathbf{x},\mathbf{y})}[\|\mathbf{y} - \mathbf{A}(\mathbf{B}\mathbf{y} + \mathbf{w})\|^2] \\
&= \mathbb{E}_{q_\phi(\mathbf{x},\mathbf{y})}[\|(\mathbf{I}_n - \mathbf{A}\mathbf{B})\mathbf{y} - \mathbf{A}\mathbf{w}\|^2] \\
&= \mathbb{E}_{q_\phi(\mathbf{x},\mathbf{y})}[[(\mathbf{I}_n - \mathbf{A}\mathbf{B})\mathbf{y} - \mathbf{A}\mathbf{w}]^\top[(\mathbf{I}_n - \mathbf{A}\mathbf{B})\mathbf{y} - \mathbf{A}\mathbf{w}]] \\
&= \mathbb{E}_{q_\phi(\mathbf{x},\mathbf{y})}[\mathrm{Tr}[[(\mathbf{I}_n - \mathbf{A}\mathbf{B})\mathbf{y} - \mathbf{A}\mathbf{w}][(\mathbf{I}_n - \mathbf{A}\mathbf{B})\mathbf{y} - \mathbf{A}\mathbf{w}]^\top]] \\
&= \mathbb{E}_{q_\phi(\mathbf{x},\mathbf{y})}[\mathrm{Tr}[[(\mathbf{I}_n - \mathbf{A}\mathbf{B})\mathbf{y} - \mathbf{A}\mathbf{w}][\mathbf{y}^\top(\mathbf{I}_n - \mathbf{A}\mathbf{B})^\top - \mathbf{w}^\top\mathbf{A}^\top]]] \\
&= \mathbb{E}_{q_\phi(\mathbf{x},\mathbf{y})}[\mathrm{Tr}[(\mathbf{I}_n - \mathbf{A}\mathbf{B})\mathbf{y}\mathbf{y}^\top(\mathbf{I}_n - \mathbf{A}\mathbf{B})^\top - (\mathbf{I}_n - \mathbf{A}\mathbf{B})\mathbf{y}\mathbf{w}^\top\mathbf{A}^\top - \mathbf{A}\mathbf{w}\mathbf{y}^\top(\mathbf{I}_n - \mathbf{A}\mathbf{B})^\top + \mathbf{A}\mathbf{w}\mathbf{w}^\top\mathbf{A}^\top]] \\
&= \mathrm{Tr}[(\mathbf{I}_n - \mathbf{A}\mathbf{B})\mathbb{E}[\mathbf{y}\mathbf{y}^\top](\mathbf{I}_n - \mathbf{A}\mathbf{B})^\top - (\mathbf{I}_n - \mathbf{A}\mathbf{B})\mathbb{E}[\mathbf{y}\mathbf{w}^\top]\mathbf{A}^\top - \mathbf{A}\mathbb{E}[\mathbf{w}\mathbf{y}^\top](\mathbf{I}_n - \mathbf{A}\mathbf{B})^\top + \mathbf{A}\mathbb{E}[\mathbf{w}\mathbf{w}^\top]\mathbf{A}^\top].
\end{aligned}
$$

The cross-covariance terms $\mathbb{E}[\mathbf{y}\mathbf{w}^\top]$ and $\mathbb{E}[\mathbf{w}\mathbf{y}^\top]$ vanish under the assumption that the observation $\mathbf{y}$ and the encoder noise $\mathbf{w}$ are independent and zero-mean, such that $\mathbb{E}[\mathbf{y}\mathbf{w}^\top] = \mathbb{E}[\mathbf{y}]\mathbb{E}[\mathbf{w}]^\top = \mathbf{0}$. Substituting the respective covariance matrices $\mathbf{\Sigma}_{\mathbf{y}}$ and $\mathbf{\Sigma}_{\mathbf{w}}$, we arrive at the final analytical form:

$$\mathbb{E}_{q_\phi(\mathbf{x},\mathbf{y})}\big[\|\mathbf{y} - \hat{\mathbf{y}}\|^2\big] = \mathrm{Tr}[(\mathbf{I}_n - \mathbf{A}\mathbf{B})\mathbf{\Sigma}_{\mathbf{y}}(\mathbf{I}_n - \mathbf{A}\mathbf{B})^\top + \mathbf{A}\mathbf{\Sigma}_{\mathbf{w}}\mathbf{A}^\top].$$

$\square$

## B Proof of Lemma 3.1

*Proof.* Following the matrix differentiation identities in Petersen & Pedersen (2006), the gradients of the objective $\mathcal{L}_\beta(\mathbf{A}, \mathbf{B}, \mathbf{\Sigma}_{\mathbf{z}}, \mathbf{\Sigma}_{\mathbf{w}})$ are computed as follows:

$$
\begin{aligned}
\frac{\partial \mathcal{L}_\beta}{\partial \mathbf{A}} &= -\frac{1}{2}\big[2\mathbf{\Sigma}_{\mathbf{z}}^{-1}\mathbf{\Sigma}_{\mathbf{y}}\mathbf{B}^\top - 2\mathbf{\Sigma}_{\mathbf{z}}^{-1}\mathbf{A}(\mathbf{B}\mathbf{\Sigma}_{\mathbf{y}}\mathbf{B}^\top + \mathbf{\Sigma}_{\mathbf{w}})\big] \\
&= \mathbf{\Sigma}_{\mathbf{z}}^{-1}\mathbf{A}(\mathbf{B}\mathbf{\Sigma}_{\mathbf{y}}\mathbf{B}^\top + \mathbf{\Sigma}_{\mathbf{w}}) - \mathbf{\Sigma}_{\mathbf{z}}^{-1}\mathbf{\Sigma}_{\mathbf{y}}\mathbf{B}^\top, \\
\frac{\partial \mathcal{L}_\beta}{\partial \mathbf{B}} &= \frac{\beta}{2}(2\mathbf{B}\mathbf{\Sigma}_{\mathbf{y}}) - \frac{1}{2}(2\mathbf{A}^\top\mathbf{\Sigma}_{\mathbf{z}}^{-1}\mathbf{\Sigma}_{\mathbf{y}} - 2\mathbf{A}^\top\mathbf{\Sigma}_{\mathbf{z}}^{-1}\mathbf{A}\mathbf{B}\mathbf{\Sigma}_{\mathbf{y}}) \\
&= \beta\mathbf{B}\mathbf{\Sigma}_{\mathbf{y}} - \mathbf{A}^\top\mathbf{\Sigma}_{\mathbf{z}}^{-1}\mathbf{\Sigma}_{\mathbf{y}} + \mathbf{A}^\top\mathbf{\Sigma}_{\mathbf{z}}^{-1}\mathbf{A}\mathbf{B}\mathbf{\Sigma}_{\mathbf{y}}, \\
\frac{\partial \mathcal{L}_\beta}{\partial \mathbf{\Sigma}_{\mathbf{w}}} &= \frac{\beta}{2}(\mathbf{I}_m - \mathbf{\Sigma}_{\mathbf{w}}^{-1}) + \frac{1}{2}\mathbf{A}^\top\mathbf{\Sigma}_{\mathbf{z}}^{-1}\mathbf{A}, \\
\frac{\partial \mathcal{L}_\beta}{\partial \mathbf{\Sigma}_{\mathbf{z}}} &= \frac{1}{2}\big[\mathbf{\Sigma}_{\mathbf{z}}^{-1}\mathbf{A}\mathbf{B}\mathbf{\Sigma}_{\mathbf{y}}\mathbf{\Sigma}_{\mathbf{z}}^{-1} + \mathbf{\Sigma}_{\mathbf{z}}^{-1}\mathbf{\Sigma}_{\mathbf{y}}\mathbf{B}^\top\mathbf{A}^\top\mathbf{\Sigma}_{\mathbf{z}}^{-1} - \mathbf{\Sigma}_{\mathbf{z}}^{-1}\mathbf{\Sigma}_{\mathbf{y}}\mathbf{\Sigma}_{\mathbf{z}}^{-1} \\
&\qquad - \mathbf{\Sigma}_{\mathbf{z}}^{-1}\mathbf{A}(\mathbf{B}\mathbf{\Sigma}_{\mathbf{y}}\mathbf{B}^\top + \mathbf{\Sigma}_{\mathbf{w}})\mathbf{A}^\top\mathbf{\Sigma}_{\mathbf{z}}^{-1} + \mathbf{\Sigma}_{\mathbf{z}}^{-1}\big].
\end{aligned}
$$

Setting $\nabla\mathcal{L}_\beta = \mathbf{0}$ yields the following first-order optimality conditions:

$$\mathbf{0} = \boldsymbol{\Sigma}_{\mathbf{z}}^{-1}\mathbf{A}(\mathbf{B}\boldsymbol{\Sigma}_{\mathbf{y}}\mathbf{B}^\top + \boldsymbol{\Sigma}_{\mathbf{w}}) - \boldsymbol{\Sigma}_{\mathbf{z}}^{-1}\boldsymbol{\Sigma}_{\mathbf{y}}\mathbf{B}^\top$$
$$\iff \mathbf{A}(\mathbf{B}\boldsymbol{\Sigma}_{\mathbf{y}}\mathbf{B}^\top + \boldsymbol{\Sigma}_{\mathbf{w}}) = \boldsymbol{\Sigma}_{\mathbf{y}}\mathbf{B}^\top$$
$$\iff \mathbf{A} = \boldsymbol{\Sigma}_{\mathbf{y}}\mathbf{B}^\top(\mathbf{B}\boldsymbol{\Sigma}_{\mathbf{y}}\mathbf{B}^\top + \boldsymbol{\Sigma}_{\mathbf{w}})^{-1}, \tag{17}$$

$$\mathbf{0} = \beta\mathbf{B}\boldsymbol{\Sigma}_{\mathbf{y}} - \mathbf{A}^\top\boldsymbol{\Sigma}_{\mathbf{z}}^{-1}\boldsymbol{\Sigma}_{\mathbf{y}} + \mathbf{A}^\top\boldsymbol{\Sigma}_{\mathbf{z}}^{-1}\mathbf{A}\mathbf{B}\boldsymbol{\Sigma}_{\mathbf{y}}$$
$$\iff (\beta\mathbf{I}_m + \mathbf{A}^\top\boldsymbol{\Sigma}_{\mathbf{z}}^{-1}\mathbf{A})\mathbf{B} = \mathbf{A}^\top\boldsymbol{\Sigma}_{\mathbf{z}}^{-1}$$
$$\iff \mathbf{B} = [\mathbf{I}_m + \mathbf{A}^\top(\boldsymbol{\Sigma}_{\mathbf{z}}^{-1}/\beta)\mathbf{A}]^{-1}\mathbf{A}^\top(\boldsymbol{\Sigma}_{\mathbf{z}}^{-1}/\beta), \tag{18}$$

$$\mathbf{0} = \boldsymbol{\Sigma}_{\mathbf{z}}^{-1}\mathbf{A}\mathbf{B}\boldsymbol{\Sigma}_{\mathbf{y}}\boldsymbol{\Sigma}_{\mathbf{z}}^{-1} + \boldsymbol{\Sigma}_{\mathbf{z}}^{-1}\boldsymbol{\Sigma}_{\mathbf{y}}\mathbf{B}^\top\mathbf{A}^\top\boldsymbol{\Sigma}_{\mathbf{z}}^{-1} - \boldsymbol{\Sigma}_{\mathbf{z}}^{-1}\boldsymbol{\Sigma}_{\mathbf{y}}\boldsymbol{\Sigma}_{\mathbf{z}}^{-1} - \boldsymbol{\Sigma}_{\mathbf{z}}^{-1}\mathbf{A}(\mathbf{B}\boldsymbol{\Sigma}_{\mathbf{y}}\mathbf{B}^\top + \boldsymbol{\Sigma}_{\mathbf{w}})\mathbf{A}^\top\boldsymbol{\Sigma}_{\mathbf{z}}^{-1} + \boldsymbol{\Sigma}_{\mathbf{z}}^{-1}$$
$$\iff \mathbf{0} = \mathbf{A}\mathbf{B}\boldsymbol{\Sigma}_{\mathbf{y}} + \boldsymbol{\Sigma}_{\mathbf{y}}\mathbf{B}^\top\mathbf{A}^\top - \boldsymbol{\Sigma}_{\mathbf{y}} - \mathbf{A}(\mathbf{B}\boldsymbol{\Sigma}_{\mathbf{y}}\mathbf{B}^\top + \boldsymbol{\Sigma}_{\mathbf{w}})\mathbf{A}^\top + \boldsymbol{\Sigma}_{\mathbf{z}}$$
$$\iff \boldsymbol{\Sigma}_{\mathbf{z}} = \boldsymbol{\Sigma}_{\mathbf{y}} - \mathbf{A}\mathbf{B}\boldsymbol{\Sigma}_{\mathbf{y}} - \boldsymbol{\Sigma}_{\mathbf{y}}\mathbf{B}^\top\mathbf{A}^\top + \mathbf{A}(\mathbf{B}\boldsymbol{\Sigma}_{\mathbf{y}}\mathbf{B}^\top + \boldsymbol{\Sigma}_{\mathbf{w}})\mathbf{A}^\top, \tag{19}$$

$$\mathbf{0} = \frac{\beta}{2}(\mathbf{I}_m - \boldsymbol{\Sigma}_{\mathbf{w}}^{-1}) + \frac{1}{2}\mathbf{A}^\top\boldsymbol{\Sigma}_{\mathbf{z}}^{-1}\mathbf{A}$$
$$\iff \beta\boldsymbol{\Sigma}_{\mathbf{w}}^{-1} = \beta\mathbf{I}_m + \mathbf{A}^\top\boldsymbol{\Sigma}_{\mathbf{z}}^{-1}\mathbf{A}$$
$$\iff \boldsymbol{\Sigma}_{\mathbf{w}} = [\mathbf{I}_m + \mathbf{A}^\top(\boldsymbol{\Sigma}_{\mathbf{z}}^{-1}/\beta)\mathbf{A}]^{-1}. \tag{20}$$

Substituting (17) into (19) and utilizing the Woodbury matrix identity (Henderson & Searle, 1981), we obtain:

$$\boldsymbol{\Sigma}_{\mathbf{z}} = \boldsymbol{\Sigma}_{\mathbf{y}} - \boldsymbol{\Sigma}_{\mathbf{y}}\mathbf{B}^\top(\mathbf{B}\boldsymbol{\Sigma}_{\mathbf{y}}\mathbf{B}^\top + \boldsymbol{\Sigma}_{\mathbf{w}})^{-1}\mathbf{B}\boldsymbol{\Sigma}_{\mathbf{y}} = (\boldsymbol{\Sigma}_{\mathbf{y}}^{-1} + \mathbf{B}^\top\boldsymbol{\Sigma}_{\mathbf{w}}^{-1}\mathbf{B})^{-1}. \tag{21}$$

Finally, we verify that the decoder gain $\mathbf{A}$ can be reformulated as $\mathbf{A} = (\boldsymbol{\Sigma}_{\mathbf{y}}^{-1} + \mathbf{B}^\top\boldsymbol{\Sigma}_{\mathbf{w}}^{-1}\mathbf{B})^{-1}\mathbf{B}^\top\boldsymbol{\Sigma}_{\mathbf{w}}^{-1}$. This form is established by verifying the following matrix identity:

$$\boldsymbol{\Sigma}_{\mathbf{y}}\mathbf{B}^\top(\mathbf{B}\boldsymbol{\Sigma}_{\mathbf{y}}\mathbf{B}^\top + \boldsymbol{\Sigma}_{\mathbf{w}})^{-1} = (\boldsymbol{\Sigma}_{\mathbf{y}}^{-1} + \mathbf{B}^\top\boldsymbol{\Sigma}_{\mathbf{w}}^{-1}\mathbf{B})^{-1}\mathbf{B}^\top\boldsymbol{\Sigma}_{\mathbf{w}}^{-1}. \tag{22}$$

To prove (22), we pre-multiply both sides by $(\boldsymbol{\Sigma}_{\mathbf{y}}^{-1} + \mathbf{B}^\top\boldsymbol{\Sigma}_{\mathbf{w}}^{-1}\mathbf{B})$ and post-multiply by $(\mathbf{B}\boldsymbol{\Sigma}_{\mathbf{y}}\mathbf{B}^\top + \boldsymbol{\Sigma}_{\mathbf{w}})$, yielding:

$$\text{LHS}: \quad (\boldsymbol{\Sigma}_{\mathbf{y}}^{-1} + \mathbf{B}^\top\boldsymbol{\Sigma}_{\mathbf{w}}^{-1}\mathbf{B})\boldsymbol{\Sigma}_{\mathbf{y}}\mathbf{B}^\top = \boldsymbol{\Sigma}_{\mathbf{y}}^{-1}\boldsymbol{\Sigma}_{\mathbf{y}}\mathbf{B}^\top + \mathbf{B}^\top\boldsymbol{\Sigma}_{\mathbf{w}}^{-1}\mathbf{B}\boldsymbol{\Sigma}_{\mathbf{y}}\mathbf{B}^\top = \mathbf{B}^\top + \mathbf{B}^\top\boldsymbol{\Sigma}_{\mathbf{w}}^{-1}\mathbf{B}\boldsymbol{\Sigma}_{\mathbf{y}}\mathbf{B}^\top,$$

$$\text{RHS}: \quad \mathbf{B}^\top\boldsymbol{\Sigma}_{\mathbf{w}}^{-1}(\mathbf{B}\boldsymbol{\Sigma}_{\mathbf{y}}\mathbf{B}^\top + \boldsymbol{\Sigma}_{\mathbf{w}}) = \mathbf{B}^\top\boldsymbol{\Sigma}_{\mathbf{w}}^{-1}\mathbf{B}\boldsymbol{\Sigma}_{\mathbf{y}}\mathbf{B}^\top + \mathbf{B}^\top\boldsymbol{\Sigma}_{\mathbf{w}}^{-1}\boldsymbol{\Sigma}_{\mathbf{w}} = \mathbf{B}^\top\boldsymbol{\Sigma}_{\mathbf{w}}^{-1}\mathbf{B}\boldsymbol{\Sigma}_{\mathbf{y}}\mathbf{B}^\top + \mathbf{B}^\top.$$

The equivalence of the transformed expressions confirms the reformulation. $\qquad\square$

## C Proof of Lemma 4.1

*Proof.* The stationary points of the $\lambda\beta$-VAE objective $\mathcal{L}_{\lambda\beta}(\mathbf{A}, \mathbf{B}, \boldsymbol{\Sigma}_{\mathbf{z}}, \boldsymbol{\Sigma}_{\mathbf{w}})$ are found by setting the gradients with respect to each parameter to zero. The gradients are derived as follows:

$$\frac{\partial\mathcal{L}_{\lambda\beta}}{\partial\mathbf{A}} = (\boldsymbol{\Sigma}_{\mathbf{z}}^{-1} + 2\lambda\mathbf{I}_n)\mathbf{A}(\mathbf{B}\boldsymbol{\Sigma}_{\mathbf{y}}\mathbf{B}^\top + \boldsymbol{\Sigma}_{\mathbf{w}}) - (\boldsymbol{\Sigma}_{\mathbf{z}}^{-1} + 2\lambda\mathbf{I}_n)\boldsymbol{\Sigma}_{\mathbf{y}}\mathbf{B}^\top, \tag{23a}$$

$$\frac{\partial\mathcal{L}_{\lambda\beta}}{\partial\mathbf{B}} = \beta\mathbf{B}\boldsymbol{\Sigma}_{\mathbf{y}} + \mathbf{A}^\top(\boldsymbol{\Sigma}_{\mathbf{z}}^{-1} + 2\lambda\mathbf{I}_n)\mathbf{A}\mathbf{B}\boldsymbol{\Sigma}_{\mathbf{y}} - \mathbf{A}^\top(\boldsymbol{\Sigma}_{\mathbf{z}}^{-1} + 2\lambda\mathbf{I}_n)\boldsymbol{\Sigma}_{\mathbf{y}}, \tag{23b}$$

$$\frac{\partial\mathcal{L}_{\lambda\beta}}{\partial\boldsymbol{\Sigma}_{\mathbf{w}}} = \frac{\beta}{2}(\mathbf{I}_m - \boldsymbol{\Sigma}_{\mathbf{w}}^{-1}) + \frac{1}{2}\mathbf{A}^\top(\boldsymbol{\Sigma}_{\mathbf{z}}^{-1} + 2\lambda\mathbf{I}_n)\mathbf{A}, \tag{23c}$$

$$\frac{\partial\mathcal{L}_{\lambda\beta}}{\partial\boldsymbol{\Sigma}_{\mathbf{z}}} = \frac{1}{2}\left[\boldsymbol{\Sigma}_{\mathbf{z}}^{-1} - \boldsymbol{\Sigma}_{\mathbf{z}}^{-1}\left(\boldsymbol{\Sigma}_{\mathbf{y}} - \mathbf{A}\mathbf{B}\boldsymbol{\Sigma}_{\mathbf{y}} - \boldsymbol{\Sigma}_{\mathbf{y}}\mathbf{B}^\top\mathbf{A}^\top + \mathbf{A}(\mathbf{B}\boldsymbol{\Sigma}_{\mathbf{y}}\mathbf{B}^\top + \boldsymbol{\Sigma}_{\mathbf{w}})\mathbf{A}^\top\right)\boldsymbol{\Sigma}_{\mathbf{z}}^{-1}\right]. \tag{23d}$$

Setting $\nabla\mathcal{L}_{\lambda\beta} = \mathbf{0}$ and defining the operator $\mathbf{M} := (\boldsymbol{\Sigma}_{\mathbf{z}}^{-1} + 2\lambda\mathbf{I}_n)/\beta$, we obtain the first-order optimality conditions. From (23a) and (23d), the decoder gain $\mathbf{A}$ and noise covariance $\boldsymbol{\Sigma}_{\mathbf{z}}$ are determined by:

$$\mathbf{A} = \boldsymbol{\Sigma}_{\mathbf{y}}\mathbf{B}^\top(\mathbf{B}\boldsymbol{\Sigma}_{\mathbf{y}}\mathbf{B}^\top + \boldsymbol{\Sigma}_{\mathbf{w}})^{-1} = (\boldsymbol{\Sigma}_{\mathbf{y}}^{-1} + \mathbf{B}^\top\boldsymbol{\Sigma}_{\mathbf{w}}^{-1}\mathbf{B})^{-1}\mathbf{B}^\top\boldsymbol{\Sigma}_{\mathbf{w}}^{-1},$$

$$\boldsymbol{\Sigma}_{\mathbf{z}} = (\boldsymbol{\Sigma}_{\mathbf{y}}^{-1} + \mathbf{B}^\top\boldsymbol{\Sigma}_{\mathbf{w}}^{-1}\mathbf{B})^{-1}.$$

Notably, these conditions are structurally identical to those of the standard $\beta$-VAE. From (23b) and (23c), the encoder gain $\mathbf{B}$ and noise covariance $\boldsymbol{\Sigma}_{\mathbf{w}}$ are coupled via the operator $\mathbf{M}$:

$$
\begin{aligned}
\mathbf{B} &= [\beta \mathbf{I}_m + \mathbf{A}^\top (\boldsymbol{\Sigma}_{\mathbf{z}}^{-1} + 2\lambda \mathbf{I}_n) \mathbf{A}]^{-1} \mathbf{A}^\top (\boldsymbol{\Sigma}_{\mathbf{z}}^{-1} + 2\lambda \mathbf{I}_n) \\
&= (\mathbf{I}_m + \mathbf{A}^\top \mathbf{M} \mathbf{A})^{-1} \mathbf{A}^\top \mathbf{M}, \\
\boldsymbol{\Sigma}_{\mathbf{w}} &= \beta [\beta \mathbf{I}_m + \mathbf{A}^\top (\boldsymbol{\Sigma}_{\mathbf{z}}^{-1} + 2\lambda \mathbf{I}_n) \mathbf{A}]^{-1} \\
&= (\mathbf{I}_m + \mathbf{A}^\top \mathbf{M} \mathbf{A})^{-1}.
\end{aligned}
$$

$\square$

# D  Proof of Theorem 3.2

This appendix provides the formal derivation of informational collapse in the $\beta$-VAE within a linear-Gaussian regime. Throughout this analysis, $\|\cdot\|_2$ denotes the spectral norm (the largest singular value), which provides the primary analytical framework for evaluating the contraction of the encoder gain. The stationarity conditions for the $\beta$-VAE objective (Lemma 3.1) define a system of coupled equations where the optimal encoder $(\mathbf{B}, \boldsymbol{\Sigma}_{\mathbf{w}})$ and decoder $(\mathbf{A}, \boldsymbol{\Sigma}_{\mathbf{z}})$ are mutually dependent. We resolve this system via the alternating optimization procedure detailed in Algorithm 1, which iteratively updates each parameter block using its closed-form stationarity condition while holding others fixed.

We demonstrate that for $\beta > 1$, this iterative sequence induces a spectral contraction of the encoder gain toward the trivial fixed point $\mathbf{B} = \mathbf{0}$. This global convergence ensures that the latent signal vanishes regardless of the initial randomized state. While we monitor the optimization process using the Frobenius norm $\|\cdot\|_F$ in our numerical implementation for computational efficiency, the equivalence of norms in finite-dimensional spaces ensures that the observed numerical behavior directly implies the spectral convergence established in our theoretical analysis.

## D.1  Uniform Spectral Bound

**Lemma D.1** (Uniform Spectral Bound). *For all iterations $t \geq 1$ of the alternating optimization procedure, the encoder noise covariance $\boldsymbol{\Sigma}_{\mathbf{w}}^{(t)}$ is spectrally bounded by unity:*

$$
\|\boldsymbol{\Sigma}_{\mathbf{w}}^{(t)}\|_2 \leq 1.
$$

*Proof.* At any iteration $t \geq 0$, the stationarity conditions established in Lemma 3.1 define the update for the subsequent state as:

$$
\boldsymbol{\Sigma}_{\mathbf{w}}^{(t+1)} = \left[ \mathbf{I}_m + \mathbf{A}^{(t)\top} \left( \boldsymbol{\Sigma}_{\mathbf{z}}^{(t)-1}/\beta \right) \mathbf{A}^{(t)} \right]^{-1}.
$$

Given that $\boldsymbol{\Sigma}_{\mathbf{z}}^{(t)} \succ \mathbf{0}$ and $\beta > 1$, the operator $\mathbf{M} := \mathbf{A}^{(t)\top} (\boldsymbol{\Sigma}_{\mathbf{z}}^{(t)-1}/\beta) \mathbf{A}^{(t)}$ is positive semi-definite ($\mathbf{M} \succeq \mathbf{0}$). It follows that $(\mathbf{I}_m + \mathbf{M}) \succeq \mathbf{I}_m$. Under the Loewner order, the matrix inverse operation reverses the inequality, yielding:

$$
\boldsymbol{\Sigma}_{\mathbf{w}}^{(t+1)} = (\mathbf{I}_m + \mathbf{M})^{-1} \preceq \mathbf{I}_m.
$$

This ordering implies that all eigenvalues $\lambda_i(\boldsymbol{\Sigma}_{\mathbf{w}}^{(t+1)})$ are contained within the interval $(0, 1]$. Consequently, the spectral norm satisfies:

$$
\|\boldsymbol{\Sigma}_{\mathbf{w}}^{(t+1)}\|_2 = \lambda_{\max}(\boldsymbol{\Sigma}_{\mathbf{w}}^{(t+1)}) \leq 1.
$$

While the initial state $\boldsymbol{\Sigma}_{\mathbf{w}}^{(0)}$ depends on the specific randomized initialization, the alternating update ensures that the sequence remains spectrally bounded for all $t \geq 1$. $\square$

---

**Algorithm 1** Alternating Optimization for $\beta$-VAE and $\lambda\beta$-VAE

---

**Require:** Data covariance $\mathbf{\Sigma_y}$, parameters $\beta, \lambda$, tolerance $\epsilon$, max iterations $T$
**Ensure:** Optimized parameters $\mathbf{A}, \mathbf{B}, \mathbf{\Sigma_z}, \mathbf{\Sigma_w}$
 1: Initialize $\mathbf{B}, \mathbf{\Sigma_w}$ randomly
 2: *flag* $\leftarrow$ True
 3: **for** $t = 0$ to $T - 1$ **do**
 4:     **if** *flag* **then**
 5:         $\mathbf{\Sigma_z} \leftarrow (\mathbf{\Sigma_y^{-1}} + \mathbf{B}^\top\mathbf{\Sigma_w^{-1}}\mathbf{B})^{-1}$
 6:         $\mathbf{A} \leftarrow \mathbf{\Sigma_z}\mathbf{B}^\top\mathbf{\Sigma_w^{-1}}$
 7:         **if** $t > 0$ and $\|\mathbf{A} - \mathbf{A}_{prev}\|_F \leq \epsilon$ and $\|\mathbf{\Sigma_z} - \mathbf{\Sigma}_{\mathbf{z},prev}\|_F \leq \epsilon$ **then**
 8:             **break**
 9:         **end if**
10:         $(\mathbf{A}_{prev}, \mathbf{\Sigma}_{\mathbf{z},prev}) \leftarrow (\mathbf{A}, \mathbf{\Sigma_z})$
11:         *flag* $\leftarrow$ False
12:     **else**
13:         **if** $\lambda = 0$ **then**
14:             $\mathbf{M} \leftarrow \mathbf{\Sigma_z^{-1}}/\beta$                                        ▷ Standard $\beta$-VAE
15:         **else**
16:             $\mathbf{M} \leftarrow (\mathbf{\Sigma_z^{-1}} + 2\lambda\mathbf{I}_n)/\beta$                              ▷ $\lambda\beta$-VAE
17:         **end if**
18:         $\mathbf{\Sigma_w} \leftarrow (\mathbf{I}_m + \mathbf{A}^\top\mathbf{M}\mathbf{A})^{-1}$
19:         $\mathbf{B} \leftarrow \mathbf{\Sigma_w}\mathbf{A}^\top\mathbf{M}$
20:         **if** $\|\mathbf{B} - \mathbf{B}_{prev}\|_F \leq \epsilon$ and $\|\mathbf{\Sigma_w} - \mathbf{\Sigma}_{\mathbf{w},prev}\|_F \leq \epsilon$ **then**
21:             **break**
22:         **end if**
23:         $(\mathbf{B}_{prev}, \mathbf{\Sigma}_{\mathbf{w},prev}) \leftarrow (\mathbf{B}, \mathbf{\Sigma_w})$
24:         *flag* $\leftarrow$ True
25:     **end if**
26: **end for**

---

## D.2 Informational Collapse

*Proof.* Let $(\mathbf{B}^{(0)}, \mathbf{\Sigma_w^{(0)}})$ denote the initial randomized state of the encoder. Following the stationarity conditions in Lemma 3.1, the corresponding optimal decoder parameters $(\mathbf{A}^{(0)}, \mathbf{\Sigma_z^{(0)}})$ are given by:

$$\mathbf{A}^{(0)} = \left[\mathbf{\Sigma_y^{-1}} + \mathbf{B}^{(0)\top}(\mathbf{\Sigma_w^{(0)}})^{-1}\mathbf{B}^{(0)}\right]^{-1}\mathbf{B}^{(0)\top}(\mathbf{\Sigma_w^{(0)}})^{-1},$$

$$\mathbf{\Sigma_z^{(0)}} = \left[\mathbf{\Sigma_y^{-1}} + \mathbf{B}^{(0)\top}(\mathbf{\Sigma_w^{(0)}})^{-1}\mathbf{B}^{(0)}\right]^{-1}.$$

Direct comparison of these expressions yields the fundamental coupling relation:

$$\mathbf{A}^{(0)} = \mathbf{\Sigma_z^{(0)}}\mathbf{B}^{(0)\top}(\mathbf{\Sigma_w^{(0)}})^{-1}. \tag{24}$$

The subsequent update for the encoder gain $\mathbf{B}^{(1)}$ in the alternating optimization procedure is:

$$\mathbf{B}^{(1)} = \left[\mathbf{I}_m + \mathbf{A}^{(0)\top}(\mathbf{\Sigma_z^{(0)-1}}/\beta)\mathbf{A}^{(0)}\right]^{-1}\mathbf{A}^{(0)\top}(\mathbf{\Sigma_z^{(0)-1}}/\beta).$$

By identifying $\mathbf{\Sigma_w^{(1)}} = [\mathbf{I}_m + \mathbf{A}^{(0)\top}(\mathbf{\Sigma_z^{(0)-1}}/\beta)\mathbf{A}^{(0)}]^{-1}$ and substituting the transpose of (24) into the expression for $\mathbf{B}^{(1)}$, we derive the recursive mapping:

$$\begin{aligned}
\mathbf{B}^{(1)} &= \beta^{-1}\mathbf{\Sigma_w^{(1)}}\mathbf{A}^{(0)\top}\mathbf{\Sigma_z^{(0)-1}} \\
&= \beta^{-1}\mathbf{\Sigma_w^{(1)}}\left[(\mathbf{\Sigma_w^{(0)}})^{-1}\mathbf{B}^{(0)}\mathbf{\Sigma_z^{(0)}}\right]\mathbf{\Sigma_z^{(0)-1}} \\
&= \beta^{-1}\mathbf{\Sigma_w^{(1)}}(\mathbf{\Sigma_w^{(0)}})^{-1}\mathbf{B}^{(0)}. \tag{25}
\end{aligned}$$

By induction, after $t$ iterations, we establish the following recursive identity for the encoder gain $\mathbf{B}^{(t)}$:

$$\mathbf{B}^{(t)} = \beta^{-t}\boldsymbol{\Sigma}_{\mathbf{w}}^{(t)}(\boldsymbol{\Sigma}_{\mathbf{w}}^{(0)})^{-1}\mathbf{B}^{(0)}. \tag{26}$$

Taking the spectral norm and utilizing the uniform bound $\|\boldsymbol{\Sigma}_{\mathbf{w}}^{(t)}\|_2 \leq 1$ for $t \geq 1$ from Lemma D.1, we obtain:

$$\|\mathbf{B}^{(t)}\|_2 \leq \beta^{-t}\|\boldsymbol{\Sigma}_{\mathbf{w}}^{(t)}\|_2 \cdot \|(\boldsymbol{\Sigma}_{\mathbf{w}}^{(0)})^{-1}\mathbf{B}^{(0)}\|_2 \leq \beta^{-t} \cdot C_0, \tag{27}$$

where $C_0 = \|(\boldsymbol{\Sigma}_{\mathbf{w}}^{(0)})^{-1}\mathbf{B}^{(0)}\|_2$ is a finite constant determined by the initialization. For $\beta > 1$, this inequality establishes that the encoder gain undergoes a spectral contraction. Consequently, the dynamics drive the encoder gain globally toward zero:

$$\|\mathbf{B}^{(t)}\|_2 \to 0 \quad \text{as } t \to \infty,$$

identifying the trivial fixed point $(\mathbf{A}, \mathbf{B}, \boldsymbol{\Sigma}_{\mathbf{z}}, \boldsymbol{\Sigma}_{\mathbf{w}}) = (\mathbf{0}, \mathbf{0}, \boldsymbol{\Sigma}_{\mathbf{y}}, \mathbf{I}_m)$ as the unique solution of the system.

To quantify the informational impact, we examine the mutual information $I(\mathbf{x}; \mathbf{v})$ using the joint covariance $\boldsymbol{\Sigma}$ defined in (4). Applying the Schur complement formula for determinants, the mutual information simplifies to:

$$I(\mathbf{x}; \mathbf{v}) = \frac{1}{2}\log\frac{\det(\boldsymbol{\Sigma}_{\mathbf{x}})}{\det(\boldsymbol{\Sigma}_{\mathbf{x}} - \mathbf{B}\boldsymbol{\Gamma}\boldsymbol{\Sigma}_{\mathbf{v}}\boldsymbol{\Gamma}^{\top}\mathbf{B}^{\top})}. \tag{28}$$

As $\mathbf{B} \to \mathbf{0}$ for $\beta > 1$, the ratio inside the logarithm approaches unity, leading to $I(\mathbf{x}; \mathbf{v}) \to 0$. This demonstrates that spectral contraction enforces a complete informational collapse, leaving the latent representation asymptotically uninformative. $\qquad\square$

## E  Proof of Informational Dynamics of $\lambda\beta$-VAE

*Proof.* Let $(\mathbf{B}^{(0)}, \boldsymbol{\Sigma}_{\mathbf{w}}^{(0)})$ denote the initial randomized state of the encoder. Following the alternating optimization procedure, we determine the corresponding optimal decoder parameters $(\mathbf{A}^{(0)}, \boldsymbol{\Sigma}_{\mathbf{z}}^{(0)})$ using the stationarity conditions established in Lemma 3.1:

$$\mathbf{A}^{(0)} = \boldsymbol{\Sigma}_{\mathbf{z}}^{(0)}\mathbf{B}^{(0)\top}(\boldsymbol{\Sigma}_{\mathbf{w}}^{(0)})^{-1}.$$

The encoder is subsequently updated to state $(\mathbf{B}^{(1)}, \boldsymbol{\Sigma}_{\mathbf{w}}^{(1)})$ utilizing the augmented stationarity conditions from Lemma 4.1. Defining the operator $\mathbf{M}^{(0)} := \beta^{-1}(\boldsymbol{\Sigma}_{\mathbf{z}}^{(0)-1} + 2\lambda\mathbf{I}_n)$, the updated encoder gain is given by:

$$\begin{aligned}
\mathbf{B}^{(1)} &= \boldsymbol{\Sigma}_{\mathbf{w}}^{(1)}\mathbf{A}^{(0)\top}\mathbf{M}^{(0)} \\
&= \beta^{-1}\boldsymbol{\Sigma}_{\mathbf{w}}^{(1)}\left[(\boldsymbol{\Sigma}_{\mathbf{w}}^{(0)})^{-1}\mathbf{B}^{(0)}\boldsymbol{\Sigma}_{\mathbf{z}}^{(0)}\right]\left(\boldsymbol{\Sigma}_{\mathbf{z}}^{(0)-1} + 2\lambda\mathbf{I}_n\right) \\
&= \beta^{-1}\boldsymbol{\Sigma}_{\mathbf{w}}^{(1)}(\boldsymbol{\Sigma}_{\mathbf{w}}^{(0)})^{-1}\mathbf{B}^{(0)}\left(\mathbf{I}_n + 2\lambda\boldsymbol{\Sigma}_{\mathbf{z}}^{(0)}\right).
\end{aligned} \tag{29}$$

By induction, after $t$ iterations, we establish the following recursive identity for the encoder gain:

$$\mathbf{B}^{(t)} = \beta^{-t}\boldsymbol{\Sigma}_{\mathbf{w}}^{(t)}(\boldsymbol{\Sigma}_{\mathbf{w}}^{(0)})^{-1}\mathbf{B}^{(0)}\prod_{k=0}^{t-1}\left(\mathbf{I}_n + 2\lambda\boldsymbol{\Sigma}_{\mathbf{z}}^{(k)}\right). \tag{30}$$

From Lemma D.1, we have $\|\boldsymbol{\Sigma}_{\mathbf{w}}^{(t)}\|_2 \leq 1$. From the optimality conditions, $\boldsymbol{\Sigma}_{\mathbf{z}} = (\boldsymbol{\Sigma}_{\mathbf{y}}^{-1} + \mathbf{B}^{\top}\boldsymbol{\Sigma}_{\mathbf{w}}^{-1}\mathbf{B})^{-1}$. Since $\mathbf{B}^{\top}\boldsymbol{\Sigma}_{\mathbf{w}}^{-1}\mathbf{B} \succeq \mathbf{0}$, it follows that $\boldsymbol{\Sigma}_{\mathbf{z}}^{-1} \succeq \boldsymbol{\Sigma}_{\mathbf{y}}^{-1}$, implying $\boldsymbol{\Sigma}_{\mathbf{z}} \preceq \boldsymbol{\Sigma}_{\mathbf{y}}$ in the Loewner order. This provides the uniform spectral bound $\|\boldsymbol{\Sigma}_{\mathbf{z}}\|_2 \leq \|\boldsymbol{\Sigma}_{\mathbf{y}}\|_2$, which yields $\|\mathbf{I}_n + 2\lambda\boldsymbol{\Sigma}_{\mathbf{z}}\|_2 \leq 1 + 2\lambda\|\boldsymbol{\Sigma}_{\mathbf{y}}\|_2$.

Applying the spectral norm and the sub-multiplicative property to (30) establishes an upper bound for the iterates:

$$\|\mathbf{B}^{(t)}\|_2 \leq \left(\frac{1 + 2\lambda\|\boldsymbol{\Sigma}_{\mathbf{y}}\|_2}{\beta}\right)^t C_0, \tag{31}$$

where $C_0 = \|(\boldsymbol{\Sigma}_{\mathbf{w}}^{(0)})^{-1}\mathbf{B}^{(0)}\|_2$ is a finite constant determined by the initialization.

For the standard $\beta$-VAE, corresponding to $\lambda = 0$, the bound reduces to $\|\mathbf{B}^{(t)}\|_2 \leq \beta^{-t} C_0$. When $\lambda > 0$, the additional factor $1 + 2\lambda\|\mathbf{\Sigma_y}\|_2$ yields a looser upper bound on encoder-gain contraction. In particular, (31) establishes exponential decay whenever $\beta > 1 + 2\lambda\|\mathbf{\Sigma_y}\|_2$. Thus, relative to the standard $\beta$-VAE, a larger value of $\beta$ is required for the derived bound to establish encoder-gain decay. This result shows that the $\lambda$-weighted reconstruction penalty weakens the derived contraction bound; it does not guarantee a non-vanishing encoder gain or, consequently, non-vanishing latent–factor mutual information.

For fixed $\lambda$, the contraction condition is eventually satisfied as $\beta$ becomes large. Specifically, the factor $\beta^{-1}(1 + 2\lambda\|\mathbf{\Sigma_y}\|_2)$ approaches zero as $\beta \to \infty$, making the derived bound increasingly contractive in the strong-regularization regime. Consequently, for sufficiently large $\beta$, $\|\mathbf{B}^{(t)}\|_2 \to 0$ as $t \to \infty$ which, by the linear-Gaussian mutual-information relation, implies $I(\mathbf{x}; \mathbf{v}) \to 0$. Thus, although the auxiliary reconstruction penalty weakens the derived contraction bound and requires a larger value of $\beta$ for the bound to establish contraction, informational collapse still occurs under sufficiently strong regularization.

$\square$

## F  Additional Reconstruction and Latent–Factor Visualizations

This appendix provides supplementary visual evaluations for the three nonlinear datasets examined in the main text: dSprites, Shapes3D, and MPI3D-real. It focuses on reconstructed observations and the mutual-information structure between latent dimensions and ground-truth generative factors. These visualizations complement the analysis in Section 5 by illustrating how reconstruction fidelity and latent–factor alignment vary across $(\beta, \lambda)$ configurations.

For each dataset, we present:

1. **Reconstruction Fidelity:** Original observations and their corresponding reconstructions, illustrating how the reconstruction quality changes with $\beta$ and $\lambda$, particularly under strong regularization.
2. **Latent–Factor Mutual Information:** Heatmaps of the mutual information between latent dimensions and ground-truth generative factors, showing how the statistical dependence structure varies across $(\beta, \lambda)$ configurations.

## G  Additional Performance Visualizations: Boxplots and Heatmaps

To complement the line plots in Figs. 3 and 5, this appendix provides additional boxplots and heatmaps for both the linear-Gaussian and deep nonlinear experiments.

- **Linear-Gaussian experiments.** The boxplots show the distributions of reconstruction error, $I(\mathbf{x}; \mathbf{v})$, SAP score, and LIM score over 100 independent optimization runs with random initialization for each $(\beta, \lambda)$ configuration under alternating optimization and AdamW. The heatmaps report the corresponding means across the full $(\beta, \lambda)$ grid, providing a compact summary of the performance landscape.
- **Deep nonlinear experiments.** The boxplots show the distributions of reconstruction error, SAP score, and MIG score over 10 independent training runs with different random seeds for each $(\beta, \lambda)$ configuration on dSprites, Shapes3D, and MPI3D-real. The heatmaps report the corresponding means across the same hyperparameter grid.

These visualizations provide complementary views of the experimental results. The line plots show how mean performance changes with $\beta$ for each fixed value of $\lambda$, the boxplots characterize variability across independent runs, and the heatmaps summarize mean performance over the full $(\beta, \lambda)$ grid.

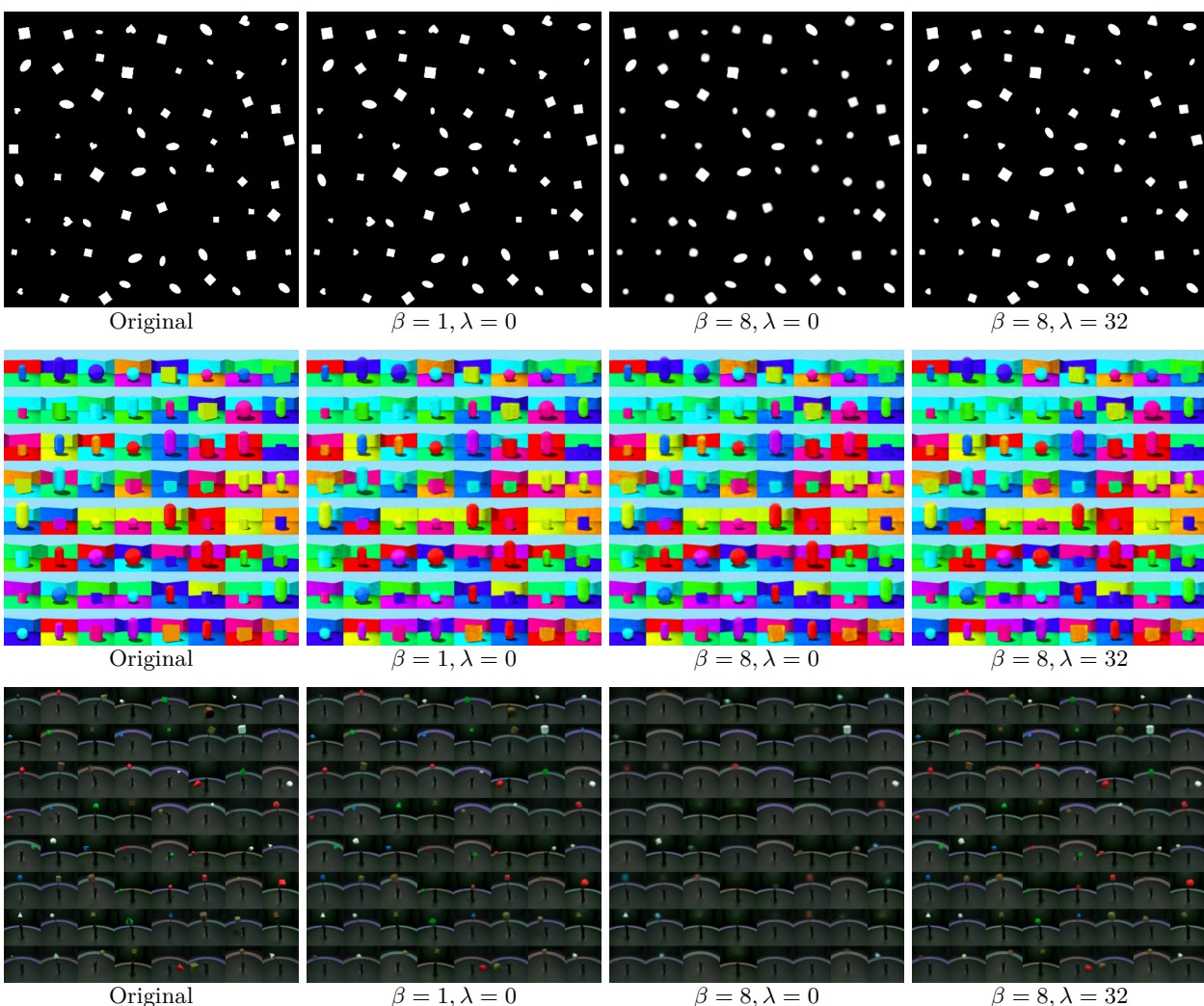

Figure 7: **Qualitative reconstructions (seed 0).** Each row shows original images alongside reconstructions for representative $(\beta, \lambda)$ configurations. For $\lambda = 0$, increasing $\beta$ produces progressively less detailed reconstructions as the latent signal weakens. At the same values of $\beta$, setting $\lambda = 32$ yields sharper reconstructions with better-preserved structural and fine-scale details.

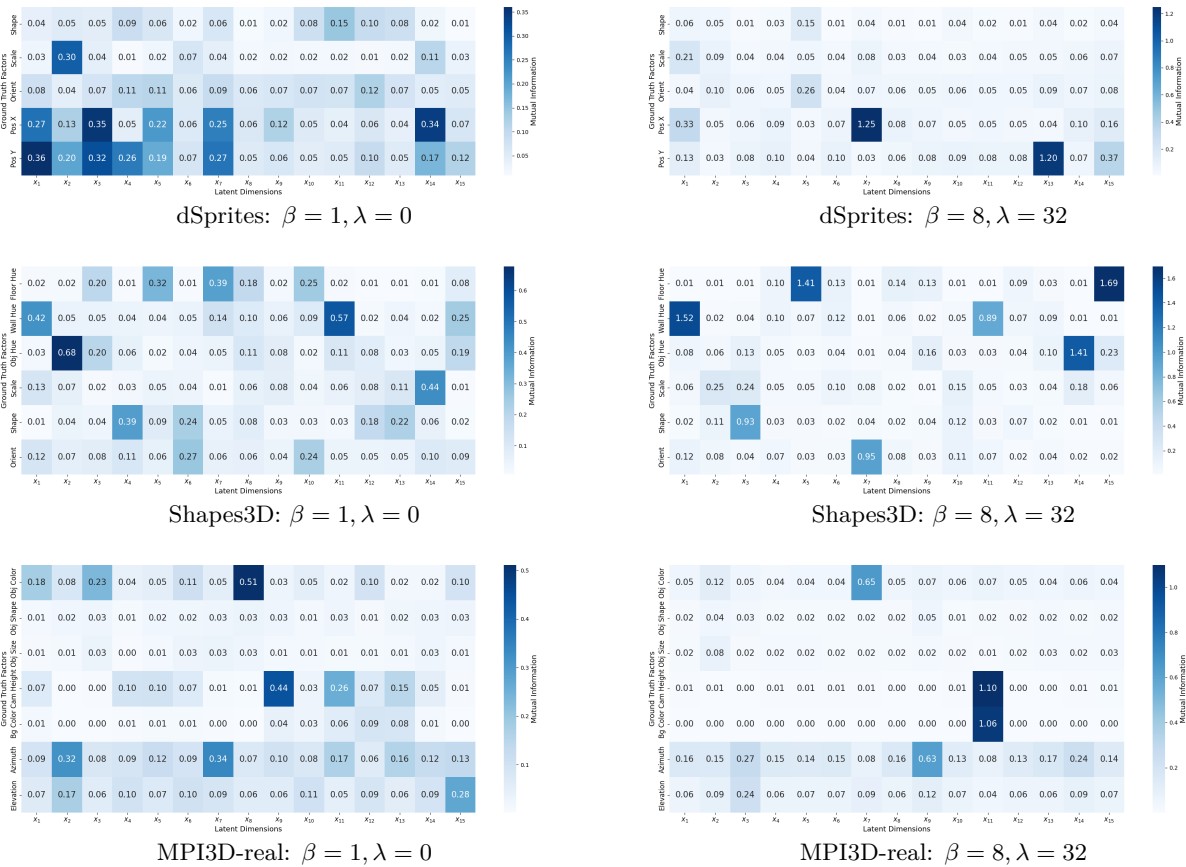

Figure 8: **Latent–factor mutual information heatmaps (seed 0).** Comparison between the low-regularization baseline ($\beta = 1, \lambda = 0$) and a more strongly regularized configuration with an auxiliary reconstruction penalty ($\beta = 8, \lambda = 32$). In this representative run, the $\lambda\beta$-VAE exhibits more localized concentrations of mutual information between particular latent dimensions and generative factors, indicating reduced redundancy and enhanced disentanglement.

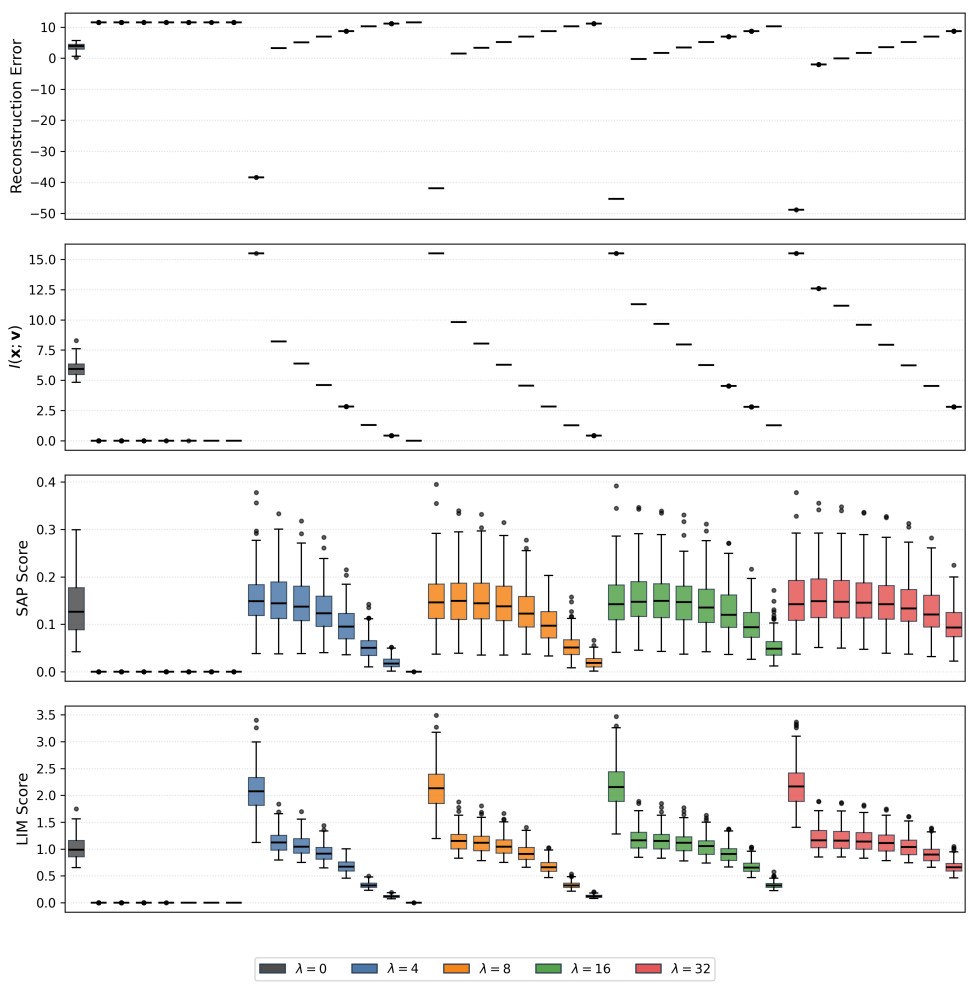

Figure 9: Boxplots of reconstruction error, $I(\mathbf{x}; \mathbf{v})$, SAP score, and LIM score for the $\lambda\beta$-VAE with alternating optimization ($n = 50, m = 10, s = 5$). Each boxplot shows the distribution of metric values over 100 independent optimization runs with random initialization for a fixed $(\beta, \lambda)$ configuration, including median, interquartile range, whiskers, and outliers. Colors encode different $\lambda$ values. For each $\lambda$, $\beta$ values are ordered from left to right as $\{1, 8, 16, 32, 64, 128, 256, 512\}$.

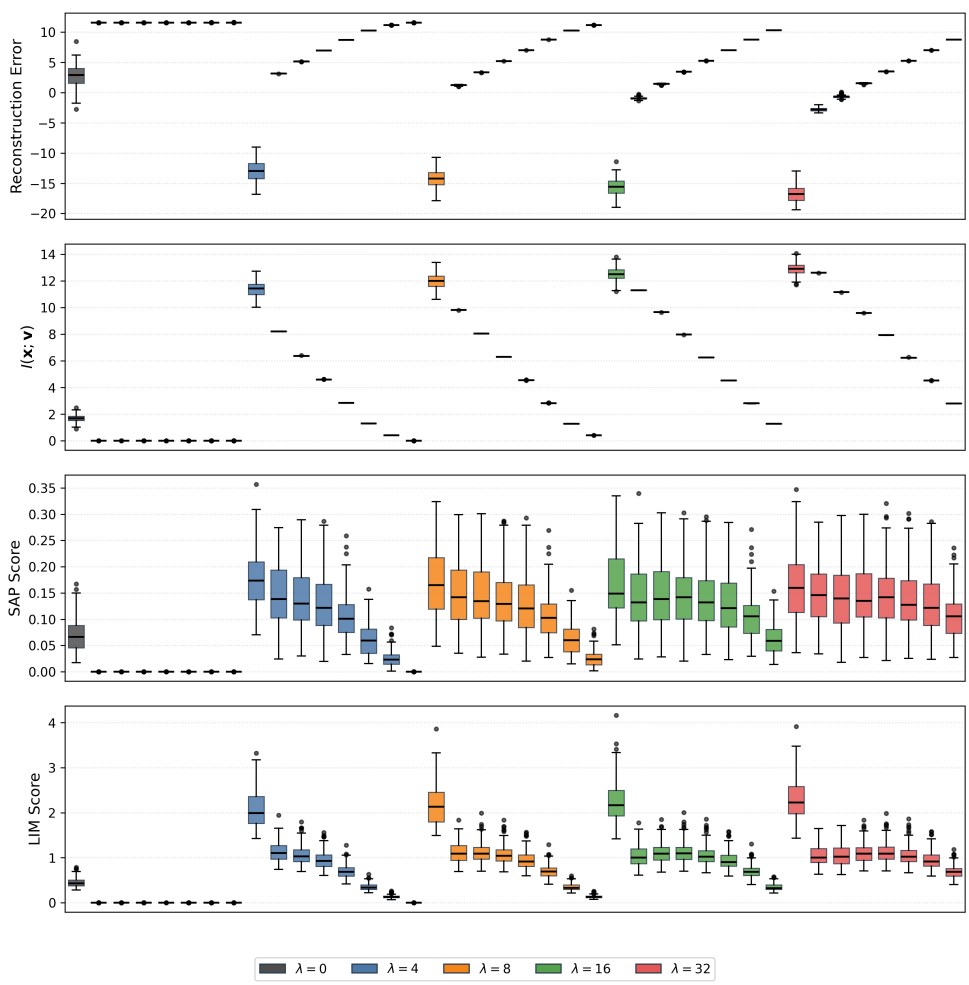

Figure 10: Boxplots of reconstruction error, $I(\mathbf{x}; \mathbf{v})$, SAP score, and LIM score for the $\lambda\beta$-VAE with AdamW optimization ($n = 50, m = 10, s = 5$). Each boxplot shows the distribution of metric values over 100 independent optimization runs with random initialization for a fixed $(\beta, \lambda)$ configuration, including median, interquartile range, whiskers, and outliers. Colors encode different $\lambda$ values. For each $\lambda$, $\beta$ values are ordered from left to right as $\{1, 8, 16, 32, 64, 128, 256, 512\}$.

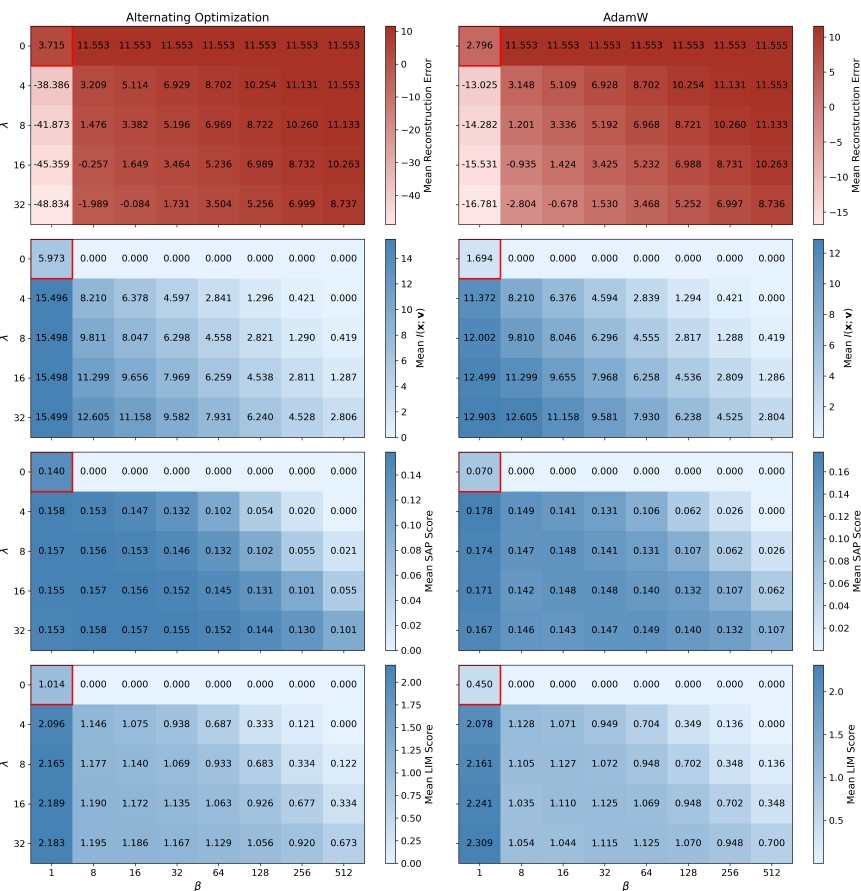

Figure 11: Heatmaps of reconstruction error, $I(\mathbf{x}; \mathbf{v})$, SAP score, and LIM score for the $\lambda\beta$-VAE ($n = 50, m = 10, s = 5$). Each cell represents the mean over 100 independent optimization runs with random initialization for a fixed $(\beta, \lambda)$ configuration.

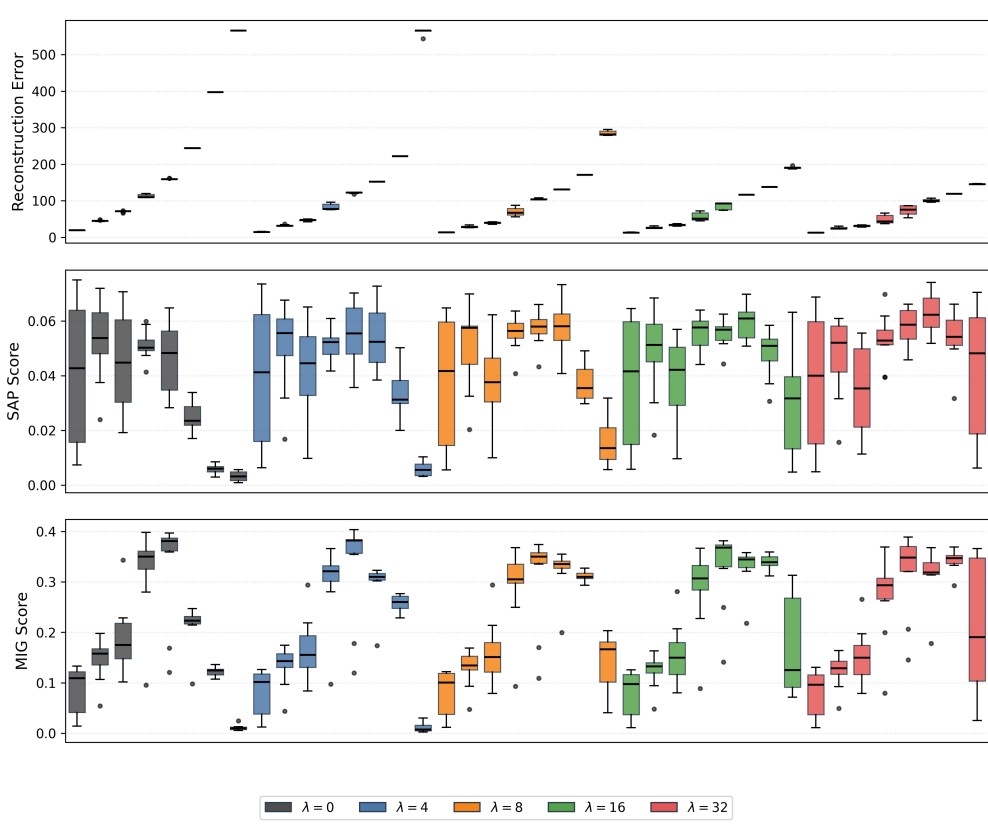

Figure 12: Boxplots of reconstruction error, SAP score, and MIG score for the $\lambda\beta$-VAE on the dSprites dataset. Each boxplot shows the distribution of metric values over 10 independent training runs with different random seeds for a fixed $(\beta, \lambda)$ configuration, including median, interquartile range, whiskers, and outliers. Colors encode different $\lambda$ values. For each $\lambda$, $\beta$ values are ordered from left to right as $\{1, 4, 8, 16, 32, 64, 128, 256\}$.

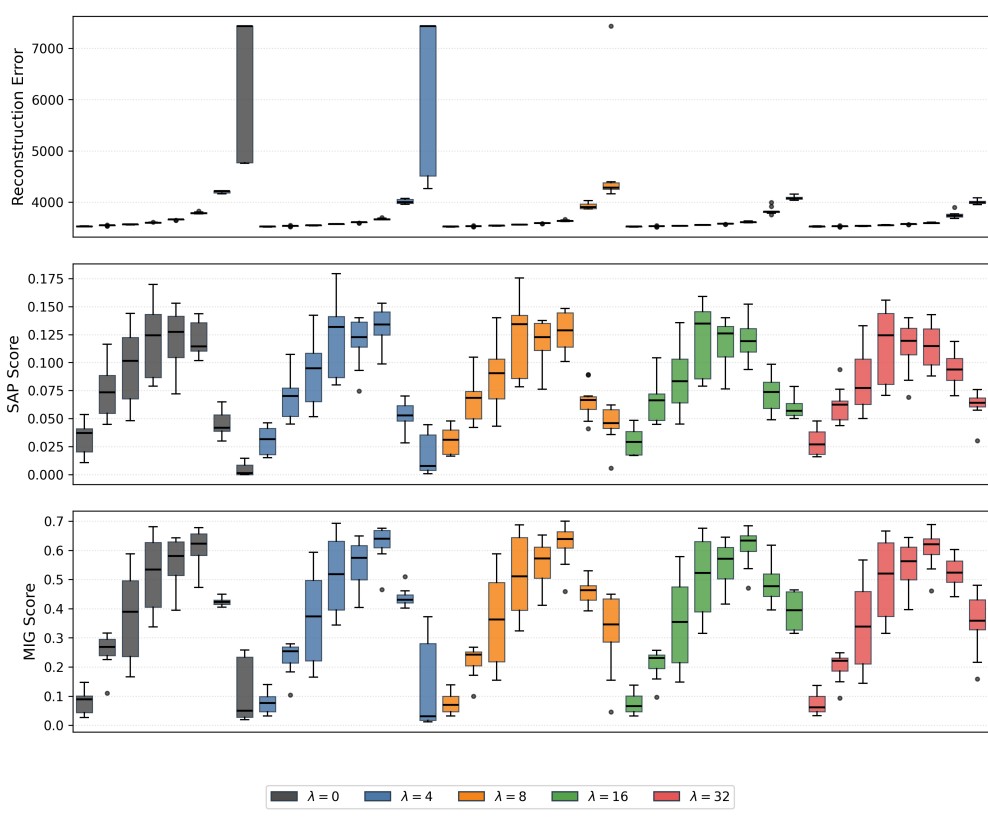

Figure 13: Boxplots of reconstruction error, SAP score, and MIG score for the $\lambda\beta$-VAE on the Shapes3D dataset. Each boxplot shows the distribution of metric values over 10 independent training runs with different random seeds for a fixed $(\beta, \lambda)$ configuration, including median, interquartile range, whiskers, and outliers. Colors encode different $\lambda$ values. For each $\lambda$, $\beta$ values are ordered from left to right as $\{1, 4, 8, 16, 32, 64, 128, 256\}$.

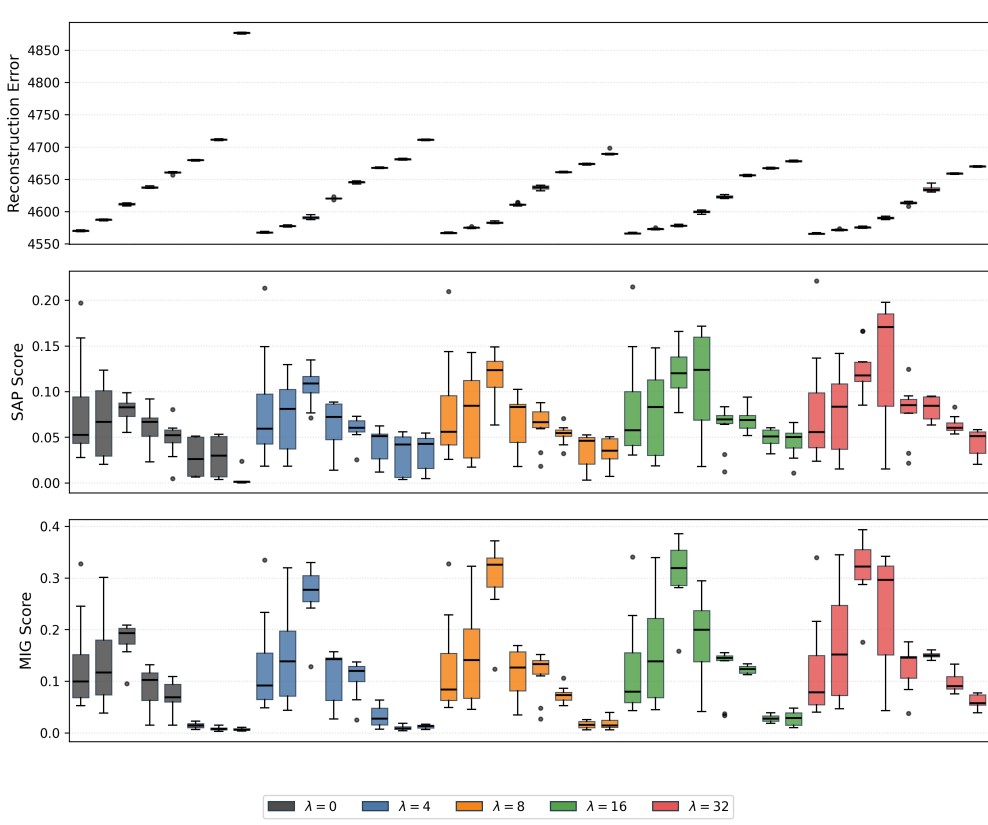

Figure 14: Boxplots of reconstruction error, SAP score, and MIG score for the $\lambda\beta$-VAE on the MPI3D-real dataset. Each boxplot shows the distribution of metric values over 10 independent training runs with different random seeds for a fixed $(\beta, \lambda)$ configuration, including median, interquartile range, whiskers, and outliers. Colors encode different $\lambda$ values. For each $\lambda$, $\beta$ values are ordered from left to right as $\{1, 4, 8, 16, 32, 64, 128, 256\}$.

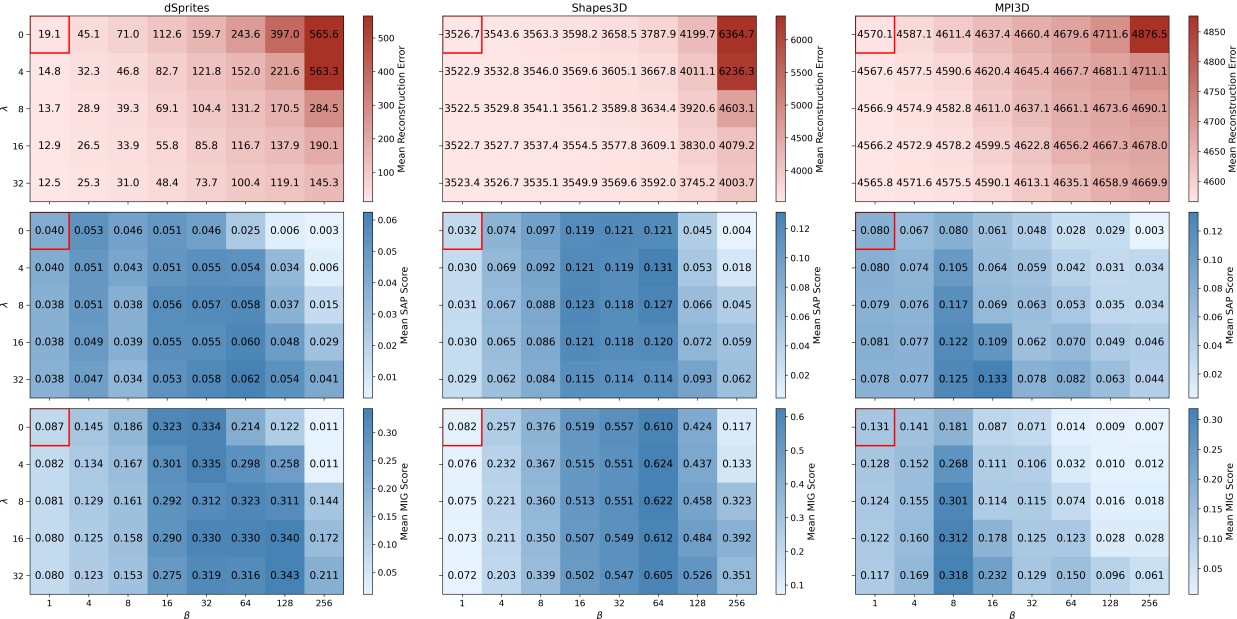

Figure 15: Heatmaps of reconstruction error, SAP score, and MIG score for the $\lambda\beta$-VAE across datasets. Each cell represents the mean over 10 independent training runs with different random seeds for a fixed $(\beta, \lambda)$ configuration.

