# OpenReview forum: "Mutual Information Collapse Explains Disentanglement Failure in $\beta$-VAEs"
_TMLR — Accepted by TMLR_

### Review · Reviewer_Q1w7 · 2026-05-07

**Summary Of Contributions:**

The paper investigates the failure of unsupervised disentanglement in $\beta$-VAEs under strong regularization. The authors theoretically characterize "informational collapse," showing that standard evaluation metrics (such as MIG and SAP) fail because excessive regularization drives the mutual information between latent variables and ground-truth generative factors toward zero. Utilizing a tractable linear-Gaussian framework, they mathematically prove that for $\beta>1$, alternating optimization induces an exponential decay in the spectral norm of the encoder gain. To mitigate this failure, the paper analyzes the $\lambda\beta$-VAE, which incorporates an auxiliary $\lambda$-weighted $L_{2}$ reconstruction penalty. Theoretical derivations and empirical evaluations on deep convolutional architectures (dSprites, Shapes3D, and MPI3D-real) demonstrate that this $\lambda$ term counteracts spectral decay, stabilizing information flow and extending the range of effective disentanglement.

## Strengths:
- Provides a strong mechanistic explanation for why disentanglement metrics like MIG and SAP break down in high-regularization regimes.
- The theoretical grounding in the linear-Gaussian model cleanly explains the empirical phenomena of spectral contraction.
- Comprehensive empirical validation confirms the restorative effects of the $\lambda$ parameter across multiple high-dimensional datasets.

## Weaknesses:
- The theoretical analysis is bounded to the linear-Gaussian regime, which, while tractable, limits direct theoretical applicability to deep non-linear architectures (though this is acknowledged as future work).
- The $\lambda$ parameter delays, but cannot entirely prevent, asymptotic informational collapse as $\beta \to \infty$.

**Audience:**

Yes

**Audience Explanation:**

Understanding the mechanics of posterior and informational collapse in VAEs is a critical and widely recognized challenge in representation learning and generative modeling. The TMLR audience, particularly those working on unsupervised disentanglement, dimensionality reduction, and generative model theory, would find the mechanistic explanation of metric failure highly relevant. Additionally, the principled justification for dual-parameter regularization ($\lambda\beta$-VAE) offers practical, actionable insights for researchers deploying these models.

**Broader Impact Concerns:**

There are no immediate ethical concerns requiring a Broader Impact Statement.

**Claims And Evidence:**

Yes

**Claims Explanation:**

The theoretical claims regarding informational collapse and the spectral contraction of the encoder gain are formally proven using the closed-form stationarity conditions within the linear-Gaussian regime. These mathematical proofs are robustly backed by controlled simulations that verify the predicted phase transitions for $\beta>1$. Furthermore, the empirical claims about the restorative effect of the $\lambda$ parameter are convincingly supported through extensive benchmarking on non-linear architectures (dSprites, Shapes3D, MPI3D-real).

**Requested Changes:**

- The paper successfully proves the mechanisms within a linear-Gaussian framework. While deep convolutional architectures are tested empirically, briefly outlining the theoretical gap between the linear-Gaussian assumptions and complex non-linear VAEs in the main text would strengthen the discussion.
- The authors rightly note that $\lambda$ does not prevent eventual informational collapse as $\beta \to \infty$. Providing more concrete heuristics or rules-of-thumb on selecting upper bounds for $\lambda$ to balance this inevitable decay in practical, non-linear scenarios would be highly beneficial for practitioners.
- The conclusion suggests dynamic scheduling of $\lambda$ as future work. A brief discussion or preliminary ablation in the appendix regarding how an adaptive $\lambda$ might compare to the static grid search methodology used  could significantly elevate the paper's practical impact. (Optional but recommended)

---

> ### Author Response · Authors · 2026-06-16
> **Response to Reviewer Q1w7**
>
> We appreciate the reviewer’s positive assessment of the paper and the thoughtful suggestions for strengthening its theoretical scope and practical interpretation. The comments helped us clarify the distinction between the linear-Gaussian analysis and the deep nonlinear experiments, refine the discussion of hyperparameter selection, and better define the scope of future work.
>
> **1. Theoretical gap between the linear-Gaussian and deep nonlinear settings.**
> We agree that the theoretical analysis is restricted to the linear-Gaussian setting under the specified alternating-optimization procedure and does not directly extend to deep nonlinear architectures. We have revised the main text to make this scope explicit. In particular, the revised manuscript states that the spectral contraction of the encoder gain and the resulting decay of latent–factor mutual information are proved only within the linear-Gaussian model. The experiments on dSprites, Shapes3D, and MPI3D-real are therefore presented as complementary empirical observations, rather than as a direct validation or theoretical extension of the linear-Gaussian result. This distinction is now emphasized in the introduction, Section 5, and conclusion.
>
> **2. Practical interpretation of the $(\beta,\lambda)$ range.**
> We agree that providing concrete heuristics or rules of thumb for selecting an upper bound on $\lambda$ in practical nonlinear settings would be highly beneficial. At present, the current results do not support a generally applicable selection rule. We have therefore revised Sections 4.2 and 5 to clarify the role of the $\lambda$-penalty and narrow the corresponding claims. In the revised manuscript, the deep nonlinear experiments show that configurations with $\lambda>0$ generally achieve lower reconstruction error over most of the evaluated $\beta$ range. Their effects on MIG and SAP are less consistent at low and moderate values of $\beta$, where the curves often overlap and no single value of $\lambda$ consistently performs best across datasets, metrics, and hyperparameter configurations. The differences are most evident in the strongly regularized regime, where configurations with $\lambda>0$ generally attain higher MIG and SAP scores than the $\lambda=0$ baseline while maintaining lower reconstruction error. We therefore interpret $\lambda$ as a useful intervention in certain strongly regularized regimes rather than as a uniquely established or universally superior mechanism. We have also reframed Section 6 as a controlled multi-objective benchmark analysis of the reconstruction–disentanglement trade-off over the evaluated $(\beta,\lambda)$ grid rather than as a practical deployment guide. Because MIG requires access to ground-truth generative factors, this analysis cannot be applied directly to hyperparameter selection in general unsupervised settings. We acknowledge the absence of a practical and generally applicable criterion for selecting $\lambda$ as a limitation of the current work. Developing such a criterion without relying on ground-truth generative factors remains an important direction for future research.
>
> **3. Dynamic scheduling of $\lambda$.**
> We thank the reviewer for suggesting a discussion or preliminary comparison between dynamic scheduling of $\lambda$ and the static grid-search methodology used in this work. In the revised manuscript, we have reframed Section 6 as a controlled multi-objective benchmark analysis of the reconstruction–disentanglement trade-off over the evaluated $(\beta,\lambda)$ grid, rather than as a practical hyperparameter-selection strategy. Dynamic tuning of $\lambda$ could provide a useful extension of the current framework. A systematic comparison with the fixed values of $\lambda$ evaluated in the static grid would require a separate study to develop an appropriate tuning criterion and assess its empirical performance. We therefore identify dynamic tuning of $\lambda$ in the revised conclusion as a promising direction for future research.

---

### Review · Reviewer_qXBw · 2026-05-20

**Summary Of Contributions:**

This paper investigates the degradation of disentanglement performance in $\beta$-VAEs under high regularization ($\beta > 1$), formally characterizing this failure mode as "informational collapse." By analyzing the stationarity conditions within a linear-Gaussian framework, the authors mathematically demonstrate that alternating optimization induces a spectral contraction of the encoder gain, which ultimately drives the mutual information between latent variables and ground-truth generative factors to zero. To counteract this decay, the paper proposes the $\lambda\beta$-VAE, which augments the standard objective with an auxiliary $L_2$ reconstruction penalty designed to stabilize information flow and preserve latent informativeness across a broader range of hyperparameter configurations.

**Key Strengths:**

* **Theoretical Clarity on standard $\beta$-VAEs:** The formalization of the informational collapse phenomenon in standard $\beta$-VAEs is a significant conceptual contribution. The linear-Gaussian proof detailing how the encoder gain undergoes spectral contraction provides a rigorous, mechanistic explanation for the widely observed empirical breakdown of disentanglement metrics (like SAP and MIG) in high-$\beta$ regimes.
* **Compelling Narrative:** The problem is highly relevant, well-motivated, and addresses a recognized bottleneck in unsupervised representation learning.

**Key Weaknesses:**

* **Theoretical Overclaims for the Proposed Method:** The mathematical proof supporting the $\lambda\beta$-VAE (Appendix E) is fundamentally incomplete regarding its primary claim. The derivation only establishes a relaxed *upper bound* on the spectral norm of the encoder gain. While this suggests that a positive $\lambda$ weakens or delays contraction, an upper bound greater than 1 does not mathematically *ensure* the prevention of informational collapse, nor does it guarantee a non-trivial fixed point.
* **Flawed Empirical Protocol:** The experimental design introduces critical confounding variables. Specifically, the two-phase training protocol grants the proposed $\lambda\beta$-VAE an additional 50,000 training steps over the baseline standard $\beta$-VAE, making it impossible to determine if improvements stem from the objective function or merely from extended training.
* **Lack of Strong Baselines:** The empirical evaluation restricts its comparisons entirely to the standard $\beta$-VAE, omitting necessary comparisons against established disentanglement architectures (e.g., $\beta$-TCVAE, FactorVAE, or DIP-VAE) that actively address the reconstruction-disentanglement trade-off. Furthermore, because $\lambda$ inherently increases reconstruction pressure, the observed results may simply reflect a lower *effective* regularization strength rather than a novel restoration mechanism.

**Additional Comments:**

N/A

**Audience:**

Yes

**Audience Explanation:**

The findings of this paper address a fundamental and widely recognized problem in unsupervised representation learning: the failure of $\beta$-VAEs to maintain informative latent spaces under strong regularization.

* **It tackles a known empirical bottleneck:** The non-monotonic behavior of disentanglement metrics (such as SAP and MIG) with respect to $\beta$ is a well-documented frustration in the field. A paper attempting to provide an analytical, root-cause explanation for this phenomenon is highly relevant.
* **It offers a useful theoretical lens:** Formalizing this failure as "informational collapse" driven by a spectral contraction of the encoder gain within a linear-Gaussian framework provides a mathematically grounded perspective on a problem that is often treated purely heuristically.

**Claims And Evidence:**

No

**Claims Explanation:**

While the paper successfully diagnoses and characterizes the informational collapse phenomenon in standard $\beta$-VAEs, the evidence supporting its proposed solution contains some theoretical and empirical flaws. The claims regarding the proposed method are not supported by convincing or mathematically accurate evidence for the following reasons:

### 1. Theoretical Overclaim (Incomplete Proof)

The authors claim that the auxiliary $\lambda$-weighted $L_2$ penalty "ensures that the encoder gain remains non-trivial even when $\beta > 1$" and prevents informational collapse. However, the mathematical proof provided in Appendix E does not support this claim.

The derivation solely establishes a relaxed **upper bound** on the spectral norm of the encoder gain when $\lambda > 0$. Proving that an upper bound can be greater than $1$ does not mathematically guarantee that the actual variable will not shrink to zero (e.g., $x_t = (0.5)^t$ satisfies the bound $x_t \le 2^t$, but still converges to zero). Without providing a lower bound, proving the existence of a non-zero fixed point, or conducting a stability analysis, the claim that the $\lambda$ intervention *ensures* the preservation of latent informativeness is a mathematical overclaim. The proof only supports that $\lambda$ might weaken or delay the contraction, not prevent it entirely.

### 2. Flawed Empirical Protocol (Confounding Variables)

The experimental design for the deep nonlinear architectures introduces a severe confounding variable that undermines the empirical claims. The authors employ a "two-phase training protocol" where the baseline $\beta$-VAE is trained for 150,000 steps, after which $\lambda$ is introduced and the model is trained for an additional 50,000 steps.

This means the proposed $\lambda\beta$-VAE receives a total of 200,000 training steps, while the baseline it is compared against only receives 150,000. Without a matched 200,000-step control group for the standard $\beta$-VAE, it is impossible to convincingly conclude whether the improvements in reconstruction fidelity and stabilized MIG/SAP metrics are due to the $\lambda$ structural intervention, or simply the result of an additional 50,000 steps of optimization.

**Requested Changes:**

### **Critical**

* **Correct the Mathematical Overclaim (Appendix E):** The authors must revise the theoretical claims surrounding the $\lambda\beta$-VAE. Currently, the text states that the $\lambda$ intervention "ensures" a non-trivial encoder gain and prevents informational collapse. However, the derivation in Appendix E only establishes a relaxed **upper bound** on the spectral norm of the encoder gain. An upper bound greater than 1 does not mathematically prevent a variable from converging to zero.


* **Establish a Fair Training Baseline:** The two-phase training protocol introduces a severe confounding variable. The proposed model is trained for 150,000 steps as a standard $\beta$-VAE, followed by 50,000 steps with the $\lambda$ penalty (200,000 steps total).
The authors are expected to include a standard $\beta$-VAE control group trained for the full 200,000 steps to definitively prove that the performance gains stem from the $\lambda$ intervention rather than simply from extended training time.


* **Isolate the Effect of Effective Regularization Strength:** Because the $\lambda$ parameter applies an additional $L_2$ reconstruction penalty, it naturally forces the model to prioritize fidelity, which could merely be acting as a reduction in the *effective* $\beta$ strength.
To prove that the $\lambda$ mechanism offers a unique structural advantage rather than just shifting the model along the standard trade-off curve, the authors should compare the $\lambda\beta$-VAE at a high $\beta$ against a standard $\beta$-VAE at a appropriately lower $\beta$ that yields a matched reconstruction error.



---

### **Would Strengthen the Work**

* **Clarify the Continuous MIG Formulation:** The standard Mutual Information Gap (MIG) normalizes by the discrete entropy of the generative factors. In the continuous linear-Gaussian setting, differential entropy can be negative or unbounded, which breaks the standard $[0, 1]$ bounding of the metric. The authors should add a brief discussion acknowledging this and explicitly frame their formulation as a continuous surrogate rather than the standard discrete MIG.

* **Acknowledge the Impracticality of the Hyperparameter Selection Strategy:** Section 6 proposes a hyperparameter selection strategy using Tchebycheff scalarization where one of the objectives is $1 - \text{MIG}$. Because MIG requires access to ground-truth generative factors, this strategy cannot be deployed in real-world unsupervised scenarios. The authors should explicitly acknowledge this limitation and reframe Section 6 as an oracle-based benchmark analysis rather than a practical deployment guide.

---

> ### Author Response · Authors · 2026-06-16
> **Response to Reviewer qXBw**
>
> We appreciate the reviewer’s detailed evaluation and thoughtful recommendations. The feedback prompted important revisions that sharpened our interpretation of the $\lambda\beta$-VAE analysis, clarified the training protocol, and more clearly defined the scope and limitations of the empirical findings.
>
> **1. Mathematical overclaim in Appendix E.**
> We agree that the original manuscript overstated what the bound in Appendix E establishes. We removed statements that the $\lambda$-weighted reconstruction penalty "ensures" latent informativeness, prevents informational collapse, or provides a restorative guarantee. The revised manuscript states that, within the linear-Gaussian setting, the auxiliary $L_2$ reconstruction penalty modifies the encoder stationarity conditions and weakens the derived upper bound on encoder-gain contraction relative to the standard $\beta$-VAE. This bound does not guarantee a non-vanishing latent–factor mutual information. For fixed $\lambda$, sufficiently large $\beta$ still makes the bound contractive and establishes informational collapse under the specified alternating-optimization procedure.
>
> **2. Fair training baseline.**
> We thank the reviewer for noting that our original description did not make the matched-compute control sufficiently explicit, creating the impression that the $\lambda=0$ baseline received a smaller training budget. For each value of $\beta$, a standard $\beta$-VAE is first trained for $150{,}000$ steps to produce a common initialization checkpoint. Training then continues for an additional $50{,}000$ steps under the $\lambda\beta$-VAE objective for each $\lambda\in$ {0,4,8,16,32}. When $\lambda=0$, the objective reduces exactly to the standard $\beta$-VAE objective. Thus, all configurations receive the same total training budget of $200{,}000$ steps. We have revised Section 5.2 to make this control explicit.
>
> **3. Effect of effective regularization strength.**
> We agree that the auxiliary $L_2$ reconstruction penalty changes the balance between reconstruction and regularization. We have narrowed the corresponding claims to the controlled linear-Gaussian experiments and the empirical behavior observed in the evaluated deep nonlinear architectures. In these settings, configurations with $\lambda>0$ can achieve lower reconstruction error and improved disentanglement performance relative to the $\lambda=0$ baseline, particularly at large values of $\beta$. In the deep nonlinear experiments, configurations with $\lambda>0$ generally achieve lower reconstruction error over most of the evaluated $\beta$ range. Their effects on MIG and SAP are less consistent at low and moderate values of $\beta$, where the curves often overlap and no single value of $\lambda$ consistently performs best. The differences are most evident in the strongly regularized regime, where configurations with $\lambda>0$ generally attain higher MIG and SAP scores while maintaining lower reconstruction error. We therefore interpret $\lambda$ as a useful intervention in certain strongly regularized regimes rather than as a uniquely established or universally superior mechanism. The current experiments do not isolate whether the observed behavior may also reflect a shift in effective regularization strength. We acknowledge this limitation and identify a reconstruction-matched comparison between a high-$\beta$ $\lambda\beta$-VAE and a lower-$\beta$ standard $\beta$-VAE as an important direction for future work.
>
> **4. Continuous MIG formulation.**
> We revised Section 4.1 to clarify that the standard MIG normalization does not transfer directly to the continuous linear-Gaussian setting. Replacing discrete entropy with differential entropy does not preserve the standard bounded normalization because differential entropy may be negative and changes under rescaling. We therefore retain MIG only for the nonlinear benchmark datasets with discrete ground-truth generative factors.
>
> **5. Practical scope of the hyperparameter-selection analysis.**
> We agree that a scalarization involving MIG cannot serve as a deployable hyperparameter-selection strategy in fully unsupervised settings because MIG requires access to ground-truth generative factors. We have therefore reframed Section 6 as a controlled multi-objective benchmark analysis of the reconstruction–disentanglement trade-off over the evaluated $(\beta,\lambda)$ grid, rather than as a practical deployment guide. The revised manuscript explicitly states that the results provide a retrospective analysis of the trade-offs observed across the tested configurations under benchmark conditions where ground-truth generative factors are available and therefore cannot be applied directly in general unsupervised settings. Developing practical hyperparameter-selection criteria that do not rely on ground-truth generative factors remains an important direction for future work.

---

### Review · Reviewer_SkSn · 2026-06-04

**Summary Of Contributions:**

The paper studies why β-VAE disentanglement is non-monotonic in the KL weight $\beta$ (MIG/SAP peak at intermediate $\beta$ and degrade under stronger regularization) and frames this as informational collapse: the vanishing of $I(x;v)$ between latents and ground-truth factors. In a linear-Gaussian model the authors derive closed-form stationarity conditions (Lemma 3.1) and prove that, under an alternating (Blahut–Arimoto-style) optimization, $\beta>1$ contracts the encoder gain to zero so $I(x;v)\to0$ (Theorem 3.2). They then study the λβ-VAE, which adds an auxiliary $L_2$ reconstruction penalty (Eq. 7), and argue it reshapes the encoder stationarity conditions (Lemma 4.1) to delay collapse. Claims are tested on synthetic data and on dSprites, Shapes3D, and MPI3D-real, with an augmented-Tchebycheff scheme for selecting $(\beta,\lambda)$.

*Strengths*:
- Targets a real problem, that strong KL regularization often destroys the latent information disentanglement relies on.
- The mathematics presented is, within its assumptions, tractable and appears sound. The collapse result follows cleanly from the recursion in Appendix D and the synthetic experiments agree.

*Weaknesses*:
- The headline claim that $\lambda$ restores informativeness is not sufficiently proven. It rests on an upper bound (Eq. 31) that is vacuous in the regime invoked.
- Conclusions are stated as general facts about β-VAEs (e.g., "inherently prone", p. 2), though the theorem holds only for the linear-Gaussian model under exact alternating updates.

**Additional Comments:**

Overall I'm concerned with a claims-to-evidence mismatch and an overextrapolation of the proved results. I would move toward acceptance if the authors correct the $\lambda$ analysis or weaken its claims to scope the conclusions to what is empirically supported, add matched-compute baselines, and report uncertainty as a way to build up confidence.

**Audience:**

Yes

**Audience Explanation:**

VAEs are an important architecture for statistics, physics, robotics, and AI (e.g. genomics, medical imaging, model explainability) and expanding the regularization theory surrounding them is of great interest to these communities. Researchers in representation learning (specifically disentanglement) and posterior collapse would value a mechanistic account of why strong KL regularization erases disentanglement meaningfulness, and they would benefit from a simple intervention that may widen the usable $\beta$ range.

**Broader Impact Concerns:**

No concerns. It's a theoretical study validated empirically on synthetic data and well-known benchmark datasets for this field

**Claims And Evidence:**

No

**Claims Explanation:**

A narrower version of the presented claims is supported, but not the full claims.

In the linear-Gaussian setting with the stated alternating updates, large $\beta$ contracts the encoder gain and eliminates $I(x;v)$ (Theorem 3.2, Appendix D). The synthetic experiments agree, and the nonlinear runs suggest the $L_2$ term can preserve reconstruction and metrics in some high-$\beta$ regimes. So, within this scope, the diagnostic claim is credible and the math is sound. But other claims, however, are stated more broadly / strongly than the presented evidence is able to support.

[1] The experiments do not isolate the mechanism. The two-phase protocol trains a β-VAE for 150k steps, then continues the same checkpoint for an additional 50k steps with $\lambda$ switched on (Table 2 and §5.2, p. 9), so each λβ-VAE accumulates 200k total steps against the 150k β-VAE baseline. Any gain therefore confounds the objective change with the extra training budget, and no matched-compute control (e.g., a β-VAE continued to 200k with $\lambda=0$) is reported. The "metric failure mode" is also largely a corollary of the collapse result (§4.1, pp. 5–6): SAP and MIG vanish as $B\to0$ because both are monotone in the latent–factor squared correlation $S_{i,j}$ (and, for MIG, in $I(x_i;v_j)=-\tfrac12\log(1-S_{i,j})$), which the contracting encoder gain drives to zero. The gap can be closed by adding evidence or narrowing the claims.

[2] The λβ "restoration" is not well-established. Appendix E bounds the gain by $\lVert B^{(t)}\rVert_2\le\big((1+2\lambda\lVert\Sigma_y\rVert_2)/\beta\big)^{t}C_0$ (Eq. 31, p. 21), an upper bound that, when its base exceeds one, merely stops proving contraction. It gives no lower bound, fixed point, or non-vanishing MI, yet the paper reads it as a guarantee ("ensuring the preservation of latent informativeness", pp. 21–22; "restorative mechanism", p. 6; "principled justification", abstract). Moreover, the collapse-prevention threshold $\beta<1+2\lambda\lVert\Sigma_y\rVert_2$ (Eq. 31) is set by the data covariance $\lVert\Sigma_y\rVert_2$, and the $L_2$ penalty is derived for a deterministic linear decoder $\hat{y}=Ax$ with the decoder noise $z$ omitted (p. 4), whereas the deep models use a Bernoulli decoder (p. 9) and show only a "more gradual" decline (p. 9); stating the result as a general fact about β-VAEs is therefore not sufficiently grounded.

As currently written I cannot answer yes, but there is a clear path to "Yes".

**Requested Changes:**

The below corrections / expansions / reframes would significantly improve my comfort in approving this paper.

[1] Fix or weaken the λβ theory: prove a lower bound / stable non-trivial fixed point for the λβ dynamics, or state that $\lambda$ only weakens the contraction bound and is >empirically< observed to delay collapse, and soften the "ensures"/"restorative"/"principled justification" language (pp. 1, 6, 7, 21–22).

[2] Separate proven from conjectured: scope the theorem to the linear-Gaussian alternating-update case, and treat general/nonlinear claims as hypotheses.

[3] Add matched-compute baselines: a β-VAE continued to the full 200k steps with $\lambda=0$ (the same +50k budget the λβ runs receive), and a from-scratch λβ-VAE at equal total compute, and reflect these in the results discussion.

[4] Report uncertainty for the λβ-VAE, not only the baselines. The standard β-VAE results already show full dispersion across runs as box plots (10 seeds for the deep models, 100 trials in the linear-Gaussian case; Figs 2 and 4), but the λβ-VAE results which carry the paper's central restoration claim are reported only as per-cell mean heatmaps (Figs 3 and 5) with no spread. Please report the same dispersion for the λβ-VAE (means with confidence intervals / error bars, or per-seed box plots) for reconstruction, MIG, SAP, and $I(x;v)$, so the restoration effect can be judged against seed variability rather than mean values alone.

---

> ### Author Response · Authors · 2026-06-16
> **Response to Reviewer SkSn**
>
> We sincerely thank the reviewer for the thoughtful and constructive assessment. The comments helped us substantially improve the precision, scope, and empirical presentation of the manuscript.
>
> **1. Overstatement of the $\lambda\beta$-VAE theory.**
> We agree that the original manuscript overstated what the bound in Appendix E establishes. We have removed statements that the $\lambda$-weighted reconstruction penalty "ensures'' latent informativeness, prevents informational collapse, or provides a restorative guarantee. The revised manuscript states that, within the linear-Gaussian setting, the auxiliary $L_2$ reconstruction penalty modifies the encoder stationarity conditions and weakens the derived upper bound on encoder-gain contraction relative to the standard $\beta$-VAE. We now clarify that this bound does not guarantee a non-vanishing encoder gain or, consequently, non-vanishing latent–factor mutual information. We also show that, for fixed $\lambda$, sufficiently large $\beta$ still makes the bound contractive and establishes informational collapse under the specified alternating-optimization procedure. The nonlinear results are now described only as empirical evidence of an attenuated decline in disentanglement performance for configurations with $\lambda>0$, particularly under strong regularization. These revisions have been incorporated throughout the abstract, introduction, Section 4.2, Section 5, Appendix E, and conclusion.
>
> **2. Scope of the theoretical claims.**
> We have restricted the theorem, its interpretation, and the surrounding conclusions to the linear-Gaussian model under the specified alternating-optimization procedure. The deep nonlinear experiments are presented as complementary empirical evidence rather than as a direct validation or extension of the theorem. These distinctions are stated explicitly in the introduction, Section 5, and conclusion.
>
> **3. Matched-compute baseline.**
> We thank the reviewer for highlighting that our original description of the training procedure did not make the matched-compute control sufficiently explicit, which may have led to the impression that the $\lambda=0$ baseline received a smaller training budget. In fact, all configurations follow a unified two-stage protocol with a total training budget of $200{,}000$ steps. For each value of $\beta$, a standard $\beta$-VAE is first trained for $150{,}000$ steps to produce a common initialization checkpoint. Starting from the corresponding pretrained checkpoint, training continues for an additional $50{,}000$ steps under the $\lambda\beta$-VAE objective for each $\lambda\in$ {0,4,8,16,32}. When $\lambda=0$, this objective reduces exactly to the standard $\beta$-VAE objective. Therefore, the $\lambda=0$ baseline receives the same total training budget as the $\lambda>0$ configurations. We have revised Section 5.2 to make this control explicit. We also thank the reviewer for suggesting a from-scratch $\lambda\beta$-VAE experiment at equal total compute. Our two-stage protocol reflects the original motivation for the $\lambda\beta$-VAE: we first train the standard $\beta$-VAE to achieve a desired level of disentanglement and then introduce the $\lambda$-weighted reconstruction penalty to reduce reconstruction error while maintaining competitive disentanglement performance, particularly at large values of $\beta$. Starting from a common pretrained checkpoint allows us to examine the effect of $\lambda$ on an already learned representation within a controlled continuation setting. The motivation for this protocol is informed by the progressive-training principle introduced in prior work [1], although our method introduces a reconstruction penalty rather than progressively increasing latent capacity. Nevertheless, we agree that an equal-compute, from-scratch comparison would provide useful complementary evidence, and we acknowledge it as a potential direction for future work.
>
> **4. Uncertainty reporting for the $\lambda\beta$-VAE.**
> We agree that mean heatmaps alone do not adequately characterize variability across runs. We therefore added uncertainty reporting for all $\lambda\beta$-VAE configurations. The main text now includes line plots presenting mean performance with 95% bootstrap confidence intervals as a function of $\beta$ for each fixed value of $\lambda$. The heatmaps have been moved to Appendix G, which also includes boxplots showing distributions across independent runs. These visualizations characterize variability in reconstruction error, $I(\mathbf{x};\mathbf{v})$, SAP, and LIM across 100 independent runs in the linear-Gaussian setting, and in reconstruction error, SAP, and MIG across 10 random seeds in the deep nonlinear experiments on dSprites, Shapes3D, and MPI3D-real.
>
>
> **Reference**
>
> [1] Christopher P. Burgess, Irina Higgins, Arka Pal, Loic Matthey, Nick Watters, Guillaume Desjardins, and Alexander Lerchner. *Understanding Disentangling in $\beta$-VAE*. 2018.

---

> > ### Comment · Reviewer_SkSn · 2026-06-18
> >
> > I thank the authors for the thorough revision, it sufficiently addressed my review points. (1) The claims are now aligned with the evidence; (2) The theoretical results are appropriately scoped; (3) The training protocol I now understand that was appropriate in the original version but was clarified in text; (4) The uncertainty quantification has been added. Consequently, I will be flipping my assessment of the manuscript to “Yes” on both criteria.
> >
> > 1. Theoretical claims for the $\lambda\beta$-VAE: The manuscript no longer states that the $\lambda$ penalty "ensures" informativeness, "prevents" collapse, or provides a "restorative"/"principled" guarantee. Section 4.2 and Appendix E now state precisely that the $L_2$ penalty only shifts the threshold at which the derived upper bound establishes contraction (from $\beta > 1$ to $\beta > 1 + 2\lambda\lVert\Sigma_y\rVert_2$). I’m comfortable now with the explicit note that this does not guarantee a non-vanishing encoder gain or non-vanishing $I(x;v)$, and that collapse still occurs for sufficiently large $\beta$.
> >
> > 2. Separating proven from: The theorem and its consequences are now restricted to the linear-Gaussian model under the specified alternating-optimization dynamics, and the deep nonlinear results are explicitly framed as complementary empirical observations. The earlier general-fact phrasing of "inherently prone" has been removed.
> >
> > 3. Matched-compute baseline: Now section 5.2 and table 2 explicit that all configurations share a 200,000-step budget, with the $\lambda = 0$ baseline continued through the full Phase 2 so that configurations differ only in $\lambda$ for each $\beta$. This clears out the training-budget confounding factor concern I had raised.
> >
> > 4. Uncertainty reporting: The authors made the three changes I asked for. First, the main-text results now report means with bootstraped confidence intervals (Figs. 3 and 5) instead of point estimates, which I feel strengthen the claim evidence. Second, the earlier mean-only heatmaps have been moved to Appendix G, where they are now accompanied by per-configuration boxplots. Third, the mutual information $I(x;v)$ is now reported directly in the linear-Gaussian setting. I checked both figures against these claims. The text now states explicitly that at low-to-moderate $\beta$ the two are closer and their confidence intervals overlap.
> > Given the revisions, the claims are now appropriately supported by the presented evidence and correctly scoped, I am therefore updating my assessment to Claims And Evidence: Yes
> >
> > I thank the authors for their thoughtful revision to their research and their openness to constructive feedback.

---

### Decision · Action_Editor_ShqJ · 2026-06-30

**Recommendation:** Accept as is

**Audience:**

Yes

**Audience Explanation:**

VAEs are a popularly used model in machine learning, but they face several issues regarding "collapse". Thus, the readership of TMLR would be interested to read a paper that may help reduce collapse for beta-VAEs/

**Claims And Evidence:**

Yes

**Claims Explanation:**

The paper investigates "informational collapse" in beta-VAEs, which occurs under excessive regularization. The authors introduce an additional L2 penalty term which can help mitigate this problem and, through theoretic analysis and empirical experimentation, they show that the new penalty term can help mitigate the occurrence of informational collapse.